# Social environment and brain structure in adolescent mental health: A cross-sectional structural equation modelling study using IMAGEN data

**Jessica Stepanous**[1]*, **Luke Munford**[2], **Pamela Qualter**[3], **Tobias Banaschewski**[4], **Frauke Nees**[4,5,6], **Rebecca Elliott**[1], **the IMAGEN Consortium**[¶]

**1** Division of Psychology and Mental Health, University of Manchester, Manchester, Greater Manchester, United Kingdom, **2** Division of Population Health, Health Services Research & Primary Care, University of Manchester, Manchester, Greater Manchester, United Kingdom, **3** Manchester Institute of Education, University of Manchester, Manchester, Greater Manchester, United Kingdom, **4** Department of Child and Adolescent Psychiatry and Psychotherapy, Central Institute of Mental Health, Medical Faculty Mannheim, Heidelberg University, Mannheim, Germany, **5** Institute of Cognitive and Clinical Neuroscience, Central Institute of Mental Health, Medical Faculty Mannheim, Heidelberg University, Mannheim, Germany, **6** Institute of Medical Psychology and Medical Sociology, University Medical Center Schleswig Holstein, Kiel University, Kiel, Germany

¶ FN and TB are part of the IMAGEN Consortium. FN is the lead IMAGEN consortium author for this publication (frauke.nees@uksh.de). The whole IMAGEN author list is provided in the Acknowledgments.
* jessica.stepanous@manchester.ac.uk

**Data Availability Statement:** Data cannot be shared publicly because they are third party data. Data are available from IMAGEN (contact via the

## Abstract

Adolescent mental health is impacted by a myriad of factors, including the developing brain, socioeconomic conditions and changing social relationships. Studies to date have neglected investigating those factors simultaneously, despite evidence of their interacting effects and distinct profiles for males and females. The current study addressed that gap by applying structural equation modelling to IMAGEN data from adolescents aged 14 years (n = 1950). A multi-group model split by sex was tested with the variables of socioeconomic stress, family support, peer problems, and brain structure as predictors, and emotional symptoms as the main outcome. Findings indicated that, for both sexes, peer problems were positively associated with emotional symptoms, and socioeconomic stress was negatively associated with family support. Additionally, there were sex-specific findings within the full models: ventromedial prefrontal cortex grey matter volume was negatively associated with emotional symptoms for males when corrected for whole brain volume, and socioeconomic stress was negatively associated with whole brain volume for females. This study underscores the importance of the peer environment for early adolescent emotional symptoms in both boys and girls, but goes further to suggest distinct gender associations with socioeconomic factors and brain structure which provides a multi-level view of risk and resilience. Future research could exploit existing IMAGEN longitudinal data to strengthen causal claims and to determine the potential longstanding impact of social environment and brain development on adolescent mental health.

IMAGEN coordinator, Jeanne Winterer jeanne.winterer@charite.de) for researchers who meet the criteria for access to confidential data. The data underlying the results presented in the study are available from IMAGEN, with more information presented on the IMAGEN website: https://imagen-project.org/?page_id=547. In order to access the IMAGEN dataset, researchers must submit a study proposal form to the IMAGEN coordinator. This will then be circulated to the IMAGEN Executive Committee, who will decide whether access will be granted to the IMAGEN dataset. If access is granted, IMAGEN will provide information on how to access the data server. We can confirm that others will be able to access the data in the same manner as the authors and the authors did not have any special access privileges that others would not have.

**Funding:** JS is funded by the ESRC-BBSRC Soc-B Centre for Doctoral Training (ES/P000347/1). TB served in an advisory or consultancy role for for eye level, Infectopharm, Lundbeck, Medice, Neurim Pharmaceuticals, Oberberg GmbH, Roche, and Takeda. He received conference support or speaker's fee by Janssen, Medice and Takeda. He received royalties from Hogrefe, Kohlhammer, CIP Medien, Oxford University Press. The present work is unrelated to the above grants and relationships. LM is partially funded by the National Institute for Health and Care Research (NIHR) Applied Research Collaboration Greater Manchester (ARC-GM; reference: NIHR200174). The views expressed in this publication are those of the author(s) and not necessarily those of the National Institute for Health and Care Research or the Department of Health and Social Care. The funders listed above provided support in the form of salaries for authors JS, TB and LM, but did not have any additional role in the study design, data collection and analysis, decision to publish, or preparation of the manuscript. The specific roles of these authors are articulated in the 'author contributions' section. The IMAGEN study was funded by the European Union-funded FP6 Integrated Project IMAGEN (Reinforcement-related behaviour in normal brain function and psychopathology) (LSHM-CT- 2007-037286), the Horizon 2020 funded ERC Advanced Grant 'STRATIFY' (Brain network based stratification of reinforcement-related disorders) (695313), Human Brain Project (HBP SGA 2, 785907, and HBP SGA 3, 945539), the Medical Research Council Grant 'c-VEDA' (Consortium on Vulnerability to Externalizing Disorders and Addictions) (MR/N000390/1), the National Institute of Health (NIH) (R01DA049238, A decentralized macro and micro gene-by-environment interaction analysis of substance use

## Introduction

Adolescent mental health is influenced by a complex, dynamic interaction of biological and social factors. One such biological factor is the structure and development of the brain, which is rapidly maturing during adolescence and refining emotional regulation abilities [1, 2]. Those processes are embedded within an increasingly complex social environment, with adolescents becoming more sensitive to peer support and exclusion [3]. Encompassing these are wider socioeconomic factors that have a top-down effect on social relationships and biological processes [4, 5]. Social and biological explanations independently provide different levels of explanation in understanding adolescent emotional symptoms, but it is clear that these levels interact to affect mental health risk and resilience [6, 7]. This is further determined by sex differences in brain development [8, 9], family support [10, 11], sensitivity to peer problems [12], and anxiety and depression symptoms [13] resulting in different pathways to mental health risk and resilience according to sex. Therefore, there is a need to consider multiple levels of explanation to obtain a comprehensive view of adolescent mental health separately for males and females. This is important as retrospective reports show that half of all individuals experiencing adult mental health conditions showed symptoms by age 14 years [14] and in the UK, 1 in 8 young people have at least one mental health problem [15]. Together, those studies show that early adolescence is a key period for individualised preventative measures and intervention. The current study provides insight into the role of both social and brain structure in adolescence for males and females separately, thus filling that gap in our understanding.

Adolescence is a time of pronounced brain development, which coincides with advances in emotional and cognitive abilities. Maturation is not uniform across the brain; there is regional variation in structural brain development across adolescence. The developmental mismatch hypothesis posits that subcortical regions mature faster than cortical regions [1, 16, 17]. This pattern of development has been used to explain the high emotional salience of peer relationships in adolescence and the resultant effect on social behaviour [17–19]. Developmental mismatch has been shown in the amygdala and prefrontal cortex (PFC), with the amygdala increasing in volume from late childhood to late adolescence (age 16 years) before stabilising in the early 20s [1, 20]. On the other hand, PFC volume decreases steadily from early adolescence into the early 20s [1]. Furthermore, these broad growth trajectories have been found to be different according to sex. For females, amygdala volume has been found to peak in early puberty; for males it has been found to increase steadily through puberty [8]. Grey matter volume in frontal regions has also been found to peak earlier in females than males, and male brain structure has been found to change more during childhood and early adolescence compared to females [9]. These dramatic changes in the adolescent brain have the potential to explain adolescence as a sensitive period for onset of mental health difficulties [17–19]. A systematic review looking at structural neuroimaging predictors of depression in childhood and adolescence found evidence for the role of reductions in prefrontal regions, however findings were not consistent. These inconsistencies were even more prevalent when looking other structures such as the amygdala [21]. One reason posited is due to a lack of consideration of sex differences in the studies. For example, one study found that onset of adolescent depression was associated with greater amygdala growth in females but attenuated growth in males between ages 12 and 16 years [6]. This reveals the importance of modelling brain development separately for males and females in adolescence, as distinct maturational profiles may be related to onset of mental health difficulties at this age.

As well as the brain, the social environment undergoes rapid development in adolescence. Adolescents begin to engage in increasingly complex social behaviours and learn to navigate the adapting social landscape with peers and family. There is evidence that males and females

behavior and its brain biomarkers), the National Institute for Health Research (NIHR) Biomedical Research Centre at South London and Maudsley NHS Foundation Trust and King's College London, the Bundesministeriumfür Bildung und Forschung (BMBF grants 01GS08152; 01EV0711; Forschungsnetz AERIAL 01EE1406A, 01EE1406B; Forschungsnetz IMAC-Mind 01GL1745B), the Deutsche Forschungsgemeinschaft (DFG grants SM 80/7-2, SFB 940, TRR 265, NE 1383/14-1), the Medical Research Foundation and Medical Research Council (grants MR/R00465X/1 and MR/S020306/1), the National Institutes of Health (NIH) funded ENIGMA (grants 5U54EB020403-05 and 1R56AG058854-01), NSFC grant 82150710554 and European Union funded project 'environMENTAL', grant no: 101057429. Further support was provided by grants from: - the ANR (ANR-12-SAMA-0004, AAPG2019 - GeBra), the Eranet Neuron (AF12-NEUR0008-01 - WM2NA; and ANR-18-NEUR00002-01 - ADORe), the Fondation de France (00081242), the Fondation pour la Recherche Médicale (DPA20140629802), the Mission Interministérielle de Lutte-contre-les-Drogues-et-les-Conduites-Addictives (MILDECA), the Assistance-Publique-Hôpitaux-de-Paris and INSERM (interface grant), Paris Sud University IDEX 2012, the Fondation de l'Avenir (grant AP-RM-17-013), the Fédération pour la Recherche sur le Cerveau; the National Institutes of Health, Science Foundation Ireland (16/ERCD/3797), U.S. A. (Axon, Testosterone and Mental Health during Adolescence; R01 MH085772-01A1) and by NIH Consortium grant U54 EB020403, supported by a cross-NIH alliance that funds Big Data to Knowledge Centres of Excellence. The funders had no role in study design, data collection and analysis, decision to publish, or preparation of the manuscript.

**Competing interests:** 'TB's commercial affiliation does not alter our adherence to PLOS ONE policies on sharing data and materials. The remaining authors have declared that no competing interests exist.'

have different perceptions of social support during adolescence, with females reporting higher levels of friend support compared to males [10, 11]. Within group, females reported receiving the most support from close friends, whilst males reported receiving the most support from parents and teachers [11]. Despite such differences in perceptions, a meta-analysis found that, in terms of the effect of support on mental health, there are more sex similarities than differences: both peer and family support have a moderate protective effect against depressive symptoms for both males and females [22]. Altogether, these studies highlight the differences in perceptions of support between adolescent males and females, but also show that the beneficial mental health effects of support exist regardless of sex. In a similar vein, poor peer relationships and peer victimisation have been found to predict depressive symptoms during adolescence in longitudinal studies [23, 24]. Whilst it is debated whether there are sex differences in the amount of peer victimisation [25], there is evidence that girls are more affected by relational victimisation than boys [12, 26, 27]. In addition, there is conflicting evidence regarding whether social support buffers against the negative effect of peer victimisation on mental health [28, 29], or whether those forms of support protect against poor mental health independently of any buffering effect [23] including only female-specific effects [30]. Thus, it is important to clarify the pathways to understand how to target interventions to improve adolescent mental health.

Social relationships are also embedded in wider contextual factors that can affect the availability and effectiveness of support. Low socioeconomic status (SES) has been widely cited as a predictor of adolescent mental health difficulties [31, 32]. One of the pathways for how SES affects mental health is through the effect on social relationships. SES has been found to negatively predict both emotional symptoms and peer problems in adolescence [33]. Additionally, SES affects the benefits of social support; the protective effect of social support against mental health difficulties has been found to be weaker in socioeconomic disadvantaged areas compared to advantaged areas [32]. SES also affects adolescent mental health through lack of parental availability, increased family stress, and reduced family support [4, 33–36]. The existence of sex differences in the relationship between SES and mental health difficulties is debated, with a systematic review finding conflicting results [31]. However, it could be argued socioeconomic status affects female mental health more than males due to their increased sensitivity to stress compared to males [35]. Therefore, it is important to disentangle the social pathways for how SES affects adolescent mental health, and whether females are more affected through the effects of stress.

The social environment has a profound impact on brain development across adolescence, which shapes risk and resilience to mental health difficulties. Young people from low income families have steeper reductions in average cortical thickness between ages 4–20 years compared to those from a high-income family [37]. In terms of family support, higher frequency of positive maternal behaviours have been found to predict attenuated growth in the right amygdala and accelerated thinning in the ventromedial/orbitofrontal cortex across early adolescence [6]. Sex-specific findings have been revealed, with neighbourhood socioeconomic disadvantage associated with greater volumetric increases in the amygdala from early to late adolescence for males but not females [7]. Positive parenting also impacts the relationship between socioeconomic disadvantage on brain development of frontal regions, and family disadvantage affects development of the amygdala in males only [7]. Taken together, it is clear that there is a nuanced relationship between sex, socioeconomic conditions, social relationships, and brain structure in mental health. The associations between socioeconomic stress, social relationships, brain structure, and mental health need to be examined for males and females separately, to determine whether there are distinct social and biological profiles for adolescent risk and resilience for males and females.

The current study addresses this gap by simultaneously modelling socioeconomic stress, social relationships–family support and peer problems–and brain structure separately for males and females. This was achieved by applying structural equation modelling to a large dataset that contains rich information on adolescent development and mental health–the IMAGEN project [38]. Cross-sectional data were selected at age 14 due to the importance of early adolescence in development of anxiety and depression symptoms [14], and due to the availability of all variables of interest at this time point. We investigated the following: how social factors interact and are associated with emotional symptoms for males and females at age 14 years, whether family support buffers against any negative effect of peer problems on mental health, how regional brain structure is associated with emotional symptoms, and whether social factors affect regional brain structure to have a cascading effect on emotional symptoms. This provided insight into the link between the social environment and brain structure, and how this affects adolescent mental health for males and females.

### Hypotheses

1. For social factors, peer problems and socioeconomic stress will positively predict emotional symptoms for both males and females at age 14 years. The effect size will be stronger for females compared to males due to the stronger negative effect of relational victimisation and stress on emotional symptoms. Socioeconomic stress will negatively predict family support, but there is no specific hypothesis about whether family support will directly predict emotional symptoms or not. In addition, no specific direction is predicted for the association between family support and peer problems, and thus whether family support mediates the relationship between peer problems and emotional symptoms.

2. There will be a significant association between amygdala and ventromedial prefrontal cortex (vmPFC) grey matter volume (GMV) and emotional symptoms, and this will be different between sex. Due to inconsistencies in the literature, no specific direction is predicted.

3. Social factors will be associated with brain structure; there will be a significant association between socioeconomic stress and amygdala/vmPFC GMV. Amygdala/vmPFC GMV will mediate the relationship between socioeconomic stress and emotional symptoms, with sex-specific findings predicted.

## Materials and methods

Data from the IMAGEN project were used. IMAGEN is a European multicentre study that contains biological, psychological, and environmental variables to assess development and behaviour in adolescence [38]. Four waves of data are available, with all participants the same age at each wave: baseline (age 14 years), follow-up 1 (age 16 years), follow-up 2 (age 19 years) and follow-up 3 (age 21 years). The current analysis uses baseline cross-sectional data at age 14 years.

### Participants

Participants were recruited from a diverse range of high schools across eight European sites (Dresden, Berlin, Mannheim, and Hamburg in Germany; London and Nottingham in the U. K.; Dublin in Ireland; and Paris in France). Only Caucasian participants were recruited for ethnic homogeneity in the genetic analysis. Written informed consent was obtained from all legal guardians. Local ethics research committees approved the study at each site, with specific information detailed in S1 Appendix in S1 File.

## Measures

The main outcome measure was emotional symptoms. Models were split by sex at age 14 years (male/female). Predictor variables included socioeconomic stress, family support, peer problems, and regional (amygdala and vmPFC) GMV. Separate latent variables were created for socioeconomic stress, family support, peer problems and emotional symptoms using the questionnaires and items presented in Table 1.

**Socioeconomic stress.** Socioeconomic stress was measured by the parent-reported socioeconomic/housing section of the Family Stresses Scale from the parent-reported Development and Well-Being Assessment (DAWBA) [39]. Parents stated the degree to which unemployment, financial difficulties, home inadequacy, and neighbour problems made family life stressful, using a three-point Likert scale.

**Family support.** Family support was measured using the affirmation section of the parent-reported Family Life Questionnaire (FLQ) [40]. Parents answered on a four-point Likert scale the degree to which their child gets love and affection, is praised and rewarded, etc.

**Peer problems.** Peer problems were measured using the peer relationship problems section of the child-reported SDQ [41]. Participants responded to items such as being alone, being liked by peers, and being bullied using a three-point Likert scale.

**Emotional symptoms.** Emotional symptoms were measured using the emotional symptoms section of the child-reported Strengths and Difficulties Questionnaire (SDQ) [41].

**Table 1. Information on the items used to construct latent variables for socioeconomic stress, family support, peer problems and emotional symptoms.**

| Latent Variable | Questionnaire | Items | Response Format |
|---|---|---|---|
| Socioeconomic Stress | Socioeconomic/Housing section of the Family Stresses Scale from the parent-reported DAWBA [38] | Do any of the following things currently make your family life stressful: • You or your partner are unemployed • Financial difficulties • Home inadequate for family's needs • Problems with neighbours/ the neighbourhood | Three-point Likert scale: • 0 = No/Does Not Apply • 1 = A little • 2 = A lot |
| Family Support | Affirmation section of the parent-reported FLQ [39] | How well do these descriptions to (child's name/your child's life) in your family? • Gets love and affection • Praised and rewarded • Gets help and support when s/he's stressed • Like and respected for who s/he is | Four-point Likert scale: • 0 = Not at all • 1 = A little • 2 = A medium amount • 3 = A great deal |
| Peer Problems | Peer Relationship Problems section of the child-reported SDQ [40] | Please give your answers on the basis of how things have been for you over the last six months: • I am usually on my own. I generally play alone or keep to myself • I have one good friend or more (negative loading) • Other people my age generally like me (negative loading) • Other children or young people pick on me or bully me • I get on better with adults than with people my own age | Three-point Likert scale: • 0 = Not True • 1 = Somewhat True • 2 = Certainly True |
| Emotional Symptoms | Emotional Symptoms section of the child-reported SDQ [40] | Please give your answers on the basis of how things have been for you over the last six months: • I get a lot of headaches, stomach-aches or sickness • I worry a lot • I am often unhappy, down-hearted or tearful • I am nervous in new situations. I easily lose confidence • I have many fears, I am easily scared | Three-point Likert scale: • 0 = Not True • 1 = Somewhat True • 2 = Certainly True |

DAWBA, Development and Well-Being Assessment; FLQ, Family Life Questionnaire; SDQ, Strengths and Difficulties Questionnaire

Participants noted the degree to which they had experienced various emotional symptoms such as somatic pains, worrying, and unhappiness in the last six months using a three-point Likert scale.

**Regional grey matter volume.**   Grey matter volume (GMV) of the amygdala and ventromedial prefrontal cortex (vmPFC) were regions of interest in the present study. Those regions were chosen due to their structural and functional significance in emotion and social relationships [42–44] and to compare potential developmental mismatch of subcortical (i.e. amygdala) compared to cortical (i.e. vmPFC) regions in adolescent brain development [1].

Structural MRI was performed on 3T scanners from different manufacturers [38]. A set of parameters was held constant across sites to address variations in image-acquisition techniques between scanners [38]. T1-weighted MR images were acquired using the magnetization prepared gradient echo sequence (MPRAGE) based on the Alzheimer's Disease Neuroimaging Initiative (ADNI) protocol [38, 45]. More details of the MR scanning protocol is described in depth elsewhere [38]. T1-weighted images were processed using FreeSurfer 5.3.0 to automatically parcellate the brain, including regional GMV. Amygdala GMV comprised left and right amygdala GMV, and was extracted using the Aseg Atlas [46]. The vmPFC was defined as the combination of left and right medial orbitofrontal cortex GMV, in line with previous studies (e.g. [47]), and extracted using the Desikan-Killiany Atlas [48].

Both uncorrected regional GMV and whole brain volume (WBV) covariate corrected GMV were explored in separate models. The WBV correction is applied to control for differences in brain size, which affects regional GMV. WBV was chosen over intracranial volume, and the covariate method was chosen over the proportionate method, because they have been found to be more reliable correction methods in developmental samples [2]. WBV was defined as the 'BrainSegVolNotVent' variable derived from FreeSurfer using the Aseg Atlas [46]. This variable contains the volume of all segmented brain regions including the cerebellum, but not including the ventricles, cerebrospinal fluid and dura [49].

**Covariates.**   Covariates in the models included psychiatric diagnosis, indicators for recruitment centre, and mean Pubertal Development Scale score. Psychiatric diagnosis was a binary variable (Yes/No) determined from any DSM-IV or ICD-10 diagnosis from the DAWBA clinical rater, who made a diagnosis from the information provided in the DAWBA [50]. Psychiatric diagnosis was added as a covariate to account for the potential effects on social, emotional, and neural measures. Recruitment centre was added as a covariate to control for potential variability in MR scanning [38]. Although the current sample are all aged 14 years, differences in pubertal status may affect factors such as brain development [51] and symptoms of anxiety and depression [52]. The Pubertal Development Scale (PDS) is a self-report measure of physical changes as a result of puberty, such as changes in height, body hair and skin, as well as male/female specific items [53]. Mean PDS scores were derived for males and females separately. Different items were available to males and females for the PDS items, such as facial hair for males and menarche for females, so these were specified accordingly. Only participants who answered all questions relevant to their sex had their mean score calculated. Exogenous categorical variables were dummy coded when entered into the model [54], which included psychiatric diagnosis (reference category = no) and recruitment centre (reference category = Berlin).

## Analysis strategy

Out of the 2315 participants with data available for any variable of interest at age 14 years, 1950 were used in the current analysis. The derivation of the sample is depicted in Fig 1. Two participants were removed from the dataset due to data quality problems identified by IMAGEN. One twin sibling was removed from the dataset; the other twin was retained.

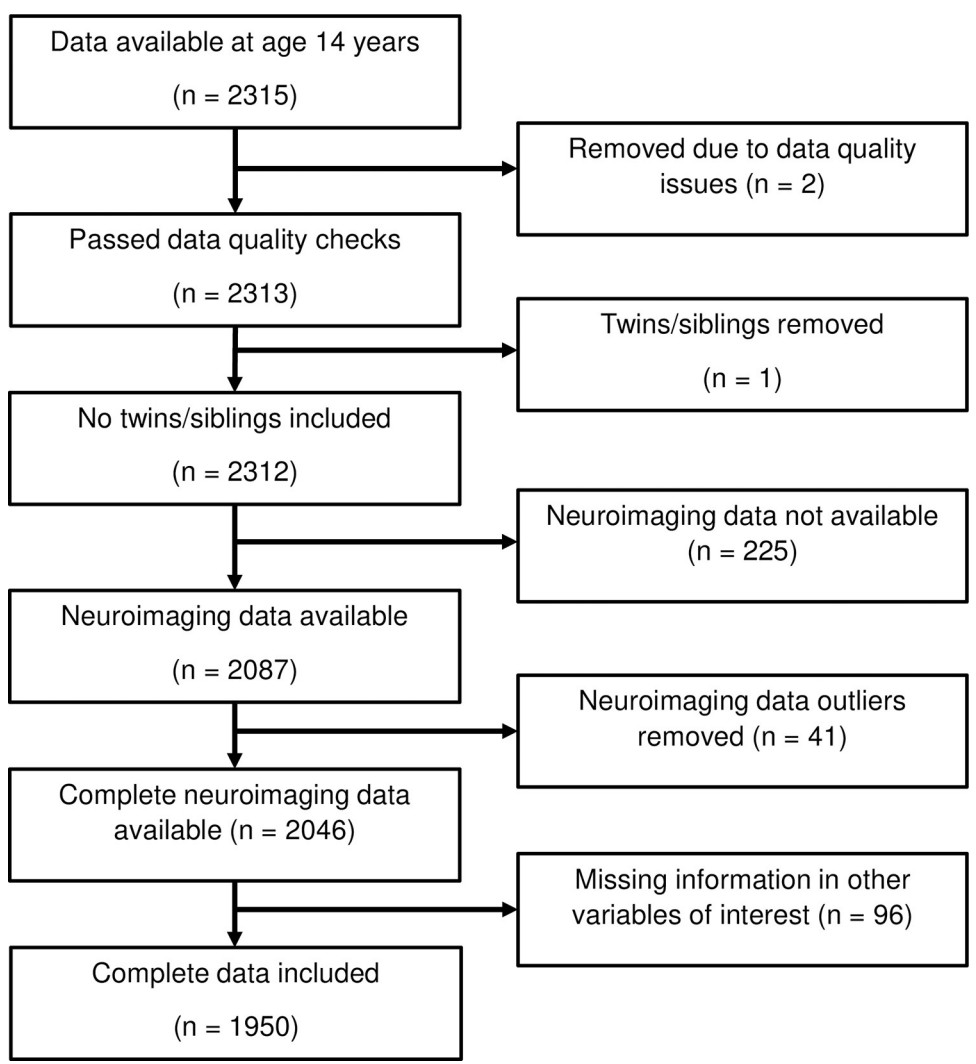

**Fig 1. Flow chart showing the derivation of the sample used for the analysis.**

The following structural equation modelling (SEM) assumptions were checked: no outliers, no missing data, and relative variances between variables [55]. Multivariate normality is typically investigated, but the current analysis included ordinal-level variables, thus weighted least squares mean- and variance-adjusted (WLSMV) estimation was used for all analyses instead of maximum likelihood (ML) estimation. WLSMV makes no assumptions about the distribution of the data andit uses diagonally weighted least squares (DWLS) to estimate the model parameters but uses the full weight matrix to calculate standard errors and a mean- and variance-adjusted test statistics [56, 57].

Univariate outliers–defined by the 'rule of thumb' of three standard deviations from the mean–were identified and removed from all neuroimaging variables to account for scanning inaccuracies and to ensure extreme values did not bias model findings. The number of outliers for each neuroimaging variable were as follows: WBV (n = 25), amygdala (n = 19), vmPFC (n = 21). Univariate and multivariate outliers were as follows: single variable (n = 21), WBV, amygdala and vmPFC (n = 4), amygdala and WBV (n = 3), vmPFC and WBV (n = 12), amygdala and vmPFC (n = 1). Multivariate outliers followed the same direction, i.e., if one value was three standard deviations below the mean, the other value also followed this.

Then, participants with complete data available in all variables used in the model were then retained for the analysis, so that there was no missing data. Ninety-six cases had data missing in the following measures: missing parent-reported data (e.g. socioeconomic stress and family support), non-completion of the Pubertal Development Scale, missing SDQ items, and missing psychiatric diagnosis information. The final sample consisted of 1950 participants. The reason for using complete data was to allow models to be run on the same data and to allow for model comparison.

In terms of relative variances between variables, amygdala GMV, vmPFC GMV and WBV were found to have variances over 1000 times larger than other variables in the model. This may be problematic as variables with large variances also have comparatively larger residual values, which means that more emphasis is placed on the larger-variance variables as the estimator calculates the parameters for the best-fitting model [58]. To address this, amygdala and vmPFC GMV values were divided by 1000, and WBV was divided by 1000000, so that the values were closer in magnitude to other variables in the model.

Next, measurement invariance analysis and structural equation modelling were conducted, with detailed information provided in the respective sections below. Analyses were conducted using the *lavaan* package (version 0.6–8) [59] in R (version 3.6.3) [60]. Measurement invariance analysis also used the *measEq.syntax* function in the *semTools* package (version 0.5–3) [61]. As mentioned previously, WLSMV estimation was used for all analyses. Model fit was assessed by the robust chi-square ($\chi^2$) fit statistic, robust root mean squared error of approximation (RMSEA) with 90% confidence interval and robust comparative fit index (CFI). Rules of thumb were used to assess model fit: robust $\chi^2$ p-value $> 0.05$, robust RMSEA $< 0.05$ and robust CFI $> 0.95$ [55] and were used as a guide rather than as strict rules. A statistically significant chi-square value is common in models with large sample sizes because there is strong statistical power to detect small differences [55]. Therefore, less emphasis was placed on this statistic.

**Measurement invariance.** Measurement invariance tests were conducted for all latent variables to assess whether the same constructs were measured for each sex. A multi-group confirmatory factor analysis was used to test sex invariance of parent-reported 'family support' and 'socioeconomic stress', and child-reported 'peer problems' and 'emotional symptoms' at age 14 years.

First, the configural model specified the structural model of the latent variables, and freely estimated the item loadings, thresholds, and residual covariance. The latent and item variables' means/intercepts were fixed to 0 and variance fixed to 1 for model identification [62]. The following constraints were then tested in sequential models: sex equivalence of item thresholds, factor loadings (metric invariance), item intercepts (scalar invariance) and residual variances (strict invariance) [62].

Equivalence of item thresholds refer to whether the boundaries between ordinal responses of an item are similar between groups. In the threshold invariance model, item thresholds are fixed to equality between groups and model fit is compared to the configural model. In order to do this, at least three degrees of freedom are required, which refers to four ordinal response categories per item [62]. This was able to be done for the family support model, however, for the socioeconomic stress, peer problems and emotional symptoms models, items only had three response categories, therefore the fit of the threshold invariance model was equivalent to the configural model due to limited degrees of freedom. For this reason, threshold invariance was assumed between sex for the socioeconomic stress, peer problems and emotional symptoms models, and this model was considered the baseline model [62]. As with the configural model, the threshold model fixed the latent variables' means/intercepts to 0 and variances to 1 for model identification; all item thresholds were fixed to equality between sex. Those

threshold restrictions allowed unnecessary identification restraints to be freed; only the reference group (female) required the item intercepts fixed to 0 and variances fixed to 1 whilst the male parameters were freely estimated [62].

Comparative model fit was assessed by comparing the fit of nested, adjacent models through changes in fit statistics (changes in CFI values ≥0.01 and RMSEA values of ≥0.015 indicate poorer fit) [63] and the scaled robust chi-square difference test statistic (significant difference indicates significantly poorer fit between models). If there were significant changes in fit, partial invariance was tested by investigating the modification indices to determine which parameter to free if it was theoretically justified. The adjusted model was than compared to the previous best-fitting model, and parameters were sequentially freed until good model fit was achieved. Individual item loadings were inspected in each CFA model. Standardised loadings at least 0.5 have practical significance [64], which was implemented as a general rule of thumb. To assess changes in model fit without low loading items, a separate model was tested which constrained the low loading item path to zero. Chi-square difference tests were conducted and differences in fit statistics, particularly CFI value, compared to determine the best fitting model. Significant differences in chi-square values favours the model with additional parameters and a higher CFI value indicates a better fitting model [65]. For comparison of factor means to be valid, equivalence of thresholds, loadings, and intercepts–also known as strong invariance–must be established at a minimum.

**Structural equation modelling.** Multi-group structural equation modelling (SEM) was used, with the analysis split by sex. A cross-sectional model of the effect of family support, peer problems, socioeconomic stress, and structural MRI measures on emotional symptoms was used (Fig 2). Model 1 contained hypothesised relationships with the uncorrected volumes for amygdala and vmPFC grey matter volume. Model 2 included WBV into the model. WBV was

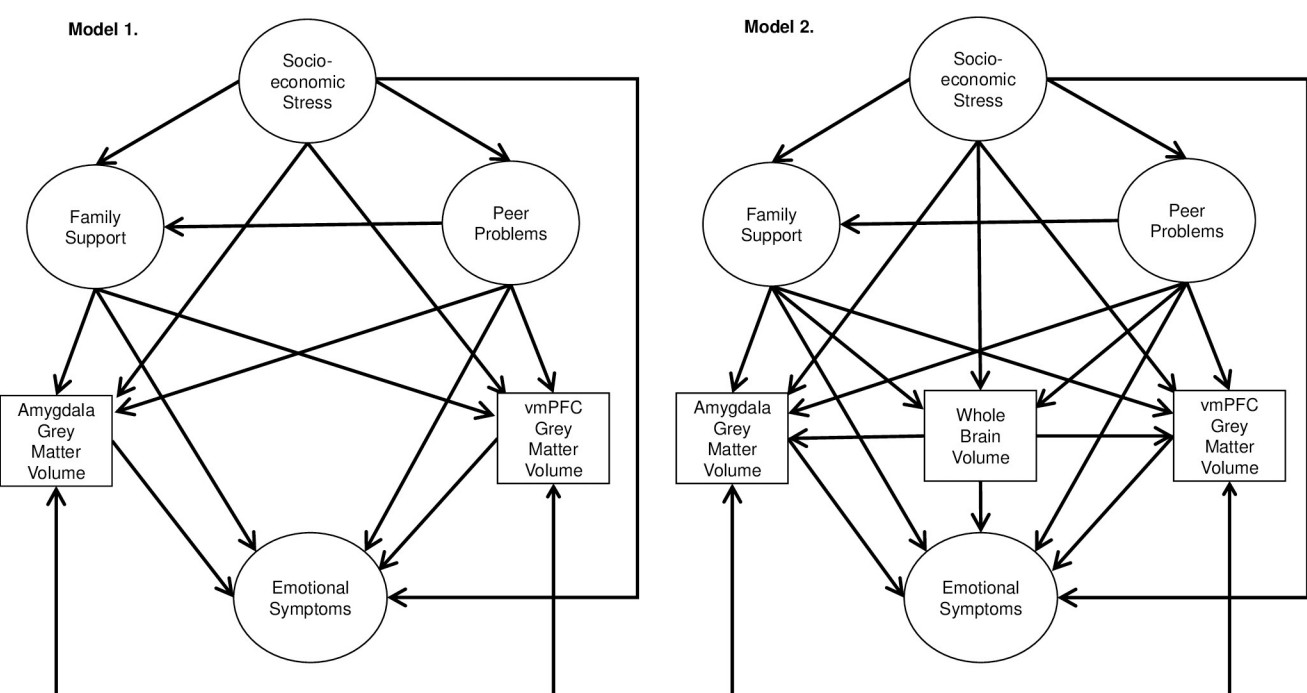

**Fig 2. Path diagrams of the structural equation models tested.** Note: Single-headed arrows show the hypothesised direction of the relationship. Double-headed arrows show covariance. Latent variables are presented in circles; observed variables are in squares. Separate models were run for males and females. Covariates and indicator variables for the latent variables are not shown for simplicity.

specified as a predictor of amygdala and vmPFC GMV to control for differences in brain size. Furthermore, WBV was specified as a predictor of emotional symptoms, and predicted by socioeconomic stress, family support and peer problems. This was to interrogate whether regional GMV associations in the model were indeed related to regional GMV or whether it was confounded by WBV. In both models, peer problems as a predictor of family support were tested to investigate the potential buffering effect of family support for peer problems and emotional symptoms. If there were a significant association between these variables, the buffering effect would be formally tested through a mediation analysis, with peer problems as the predictor, emotional symptoms as the outcome and family support as the mediator.

First, model fit was assessed for each model individually. Then, to allow for model comparison, model 1 was nested within model 2 by fixing paths not present in the model (i.e., those including WBV) to zero. Nested models were compared using chi-square difference tests and comparing improvements in other fit statistics, such as CFI values [65].

## Results and discussion

### Descriptive statistics

Descriptive statistics for the continuous variables in the sample with complete data are shown in Table 2. There were slightly more females (n = 1001) than males (n = 949) in the sample, however this difference was not significant ($\chi^2$ = 1.387, df = 1, p = 0.239). The mean Pubertal Development Score was significantly greater for females compared to males. As expected, whole brain volume, amygdala and vmPFC GMV were significantly larger on average in males compared to females. Furthermore, amygdala and vmPFC GMV had a larger standard deviation in males compared to females.

Responses to categorical and ordinal-level items are detailed in Table 3. A higher proportion of females had a psychiatric diagnosis compared to males ($\chi^2$ = 5.945, df = 1, p = 0.015). Recruitment was fairly distributed; Dublin had a smaller proportion and Nottingham had a larger proportion of the sample, but this was the same for both sexes ($\chi^2$ = 5.528, df = 7, p = 0.596). Most parents positively affirmed family support items. However, for the item "Liked and respected for who s/he is", there was a significant sex difference ($\chi^2$ = 9.018, df = 3, p = 0.029). Parents of male adolescents were more likely to respond "A medium amount" (post-hoc residual = 2.994, p = 0.022) and less likely to respond "A great deal" (post-hoc residual = -2.811, p = 0.040) compared to parents of female adolescents. There were sex differences

**Table 2. Descriptive statistics for the continuous variables in the sample with complete data, separately for males and females (N = 1950).**

| Variable | Males (n = 949) | | | | | Females (n = 1001) | | | | | Sex difference Welch Two Sample t-test |
| --- | --- | --- | --- | --- | --- | --- | --- | --- | --- | --- | --- |
| | Mean (SD) | Min | Max | Skew | Kurtosis | Mean (SD) | Min | Max | Skew | Kurtosis | t (df) |
| **Mean PDS Score** | 2.60 (0.53) | 1.0 | 4.0 | -0.48 | 0.13 | 3.19 (0.43) | 1.4 | 4.0 | -0.83 | 1.03 | 27.102 (1820.9)*** |
| **WBV ($mm^3$)** | 1230047.60 (107322.70) | 797281.0 | 1528026.0 | -0.57 | 1.38 | 1108494.43 (93532.08) | 789834.0 | 1468714.0 | -0.09 | 0.40 | -26.603 (1880.5)*** |
| **Amygdala ($mm^3$)** | 3739.11 (437.80) | 2105.7 | 5036.4 | -0.11 | 0.14 | 3381.47 (414.38) | 2136.4 | 4992.8 | 0.15 | 0.05 | -18.505 (1925.5)*** |
| **vmPFC ($mm^3$)** | 11875.98 (1451.97) | 6560.0 | 16035.0 | -0.18 | 0.30 | 10840.78 (1297.87) | 6887.0 | 15467.0 | 0.09 | 0.01 | -16.567 (1896.5)*** |

Note.

*** = means are statistically significantly different between sex, p < .001. PDS, Pubertal Development Scale; vmPFC, ventromedial prefrontal cortex; WBV, whole brain volume

**Table 3. Count data for the categorical and ordinal variables separately for males and females, expressed as both frequency and row percentage.**

| Variables | Males (n = 949) | | | | Females (n = 1001) | | | |
|---|---|---|---|---|---|---|---|---|
| | Response Options | | | | Response Options | | | |
| **Psychiatric Diagnosis** | **Yes** | **No** | | | **Yes** | **No** | | |
| | 105 (11.06%) | 844 (88.94%) | | | 149 (14.89%) | 852 (85.11%) | | |
| **Recruitment Centre** | **Berlin** | **Dresden** | **Dublin** | **Hamburg** | **Berlin** | **Dresden** | **Dublin** | **Hamburg** |
| | 114 (12.01%) | 124 (13.07%) | 94 (9.91%) | 111 (11.70%) | 132 (13.19%) | 120 (11.99%) | 87 (8.69%) | 134 (13.39%) |
| | **London** | **Mannheim** | **Nottingham** | **Paris** | **London** | **Mannheim** | **Nottingham** | **Paris** |
| | 109 (11.49%) | 100 (10.54%) | 171 (18.02%) | 126 (13.28%) | 130 (12.99%) | 114 (11.39%) | 161 (16.08%) | 123 (12.29%) |
| **Family Support Indicators** | **Not at all** | **A little** | **A medium amount** | **A great deal** | **Not at all** | **A little** | **A medium amount** | **A great deal** |
| Gets love and affection | 1 (0.11%) | 71 (7.48%) | 382 (40.25%) | 495 (52.16%) | 1 (0.10%) | 68 (6.79%) | 390 (38.96%) | 542 (54.15%) |
| Praised and rewarded | 1 (0.11%) | 18 (1.90%) | 175 (18.44%) | 755 (79.56%) | 2 (0.20%) | 17 (1.70%) | 160 (15.98%) | 822 (82.12%) |
| Gets help and support when s/he's stressed | 5 (0.53%) | 34 (3.58%) | 198 (20.86%) | 712 (75.03%) | 6 (0.60%) | 32 (3.20%) | 183 (18.28%) | 780 (77.92%) |
| Liked and respected for who s/he is | 2 (0.21%) | 21 (2.21%) | 149 (15.70%) | 777 (81.88%) | 2 (0.20%) | 22 (2.20%) | 111 (11.09%) | 866 (86.51%) |
| **Emotional Symptoms Indicators** | **Not true** | **Somewhat true** | **Certainly true** | | **Not true** | **Somewhat true** | **Certainly true** | |
| I get a lot of headaches, stomach-aches or sickness | 686 (72.29%) | 211 (22.23%) | 52 (5.48%) | | 529 (52.85%) | 382 (38.16%) | 90 (8.99%) | |
| I worry a lot | 461 (48.58%) | 384 (40.46%) | 104 (10.96%) | | 284 (28.37%) | 484 (48.35%) | 233 (23.28%) | |
| I am often unhappy, down-hearted or tearful | 787 (82.93%) | 139 (14.65%) | 23 (2.42%) | | 605 (60.44%) | 330 (32.97%) | 66 (6.59%) | |
| I am nervous in new situations. I easily lose confidence | 507 (53.42%) | 345 (36.35%) | 97 (10.22%) | | 338 (33.77%) | 466 (46.55%) | 197 (19.68%) | |
| I have many fears, I am easily scared | 746 (78.61%) | 185 (19.49%) | 18 (1.90%) | | 598 (59.74%) | 343 (34.27%) | 60 (5.99%) | |
| **Peer Problems Indicators** | **Not true** | **Somewhat true** | **Certainly true** | | **Not true** | **Somewhat true** | **Certainly true** | |
| I am usually on my own. I generally play alone or keep to myself | 552 (58.17%) | 321 (33.83%) | 76 (8.01%) | | 609 (60.84%) | 334 (33.37%) | 58 (5.79%) | |
| I have one good friend or more (negative loading) | 16 (1.69%) | 91 (9.59%) | 842 (88.72%) | | 9 (0.90%) | 61 (6.09%) | 931 (93.01%) | |
| Other people my age generally like me (negative loading) | 44 (4.64%) | 432 (45.52%) | 473 (49.84%) | | 40 (4.00%) | 463 (46.25%) | 498 (49.75%) | |
| Other children or young people pick on me or bully me | 778 (81.98%) | 138 (14.54%) | 33 (3.48%) | | 852 (85.11%) | 122 (12.19%) | 27 (2.70%) | |
| I get on better with adults than with people my own age | 540 (56.90%) | 348 (36.67%) | 61 (6.43%) | | 596 (59.54%) | 342 (34.17%) | 63 (6.29%) | |
| **Socioeconomic Stress Indicators** | **No/Does not apply** | **A little** | **A lot** | | **No/Does not apply** | **A little** | **A lot** | |
| You or your partner are unemployed | 861 (90.73%) | 56 (5.90%) | 32 (3.37%) | | 889 (88.81%) | 79 (7.89%) | 33 (3.30%) | |
| Financial difficulties | 642 (67.65%) | 248 (26.13%) | 59 (6.22%) | | 665 (66.43%) | 273 (27.27%) | 63 (6.29%) | |
| Home inadequate for family's needs | 854 (89.99%) | 75 (7.90%) | 20 (2.11%) | | 891 (89.01%) | 98 (9.79%) | 12 (1.20%) | |
| Problems with neighbours/ the neighbourhood | 891 (93.89%) | 54 (5.69%) | 4 (0.42%) | | 950 (94.91%) | 43 (4.30%) | 8 (0.80%) | |

in responses to all emotional symptoms items (all $\chi^2 \geq 78.436$, df = 2, ps < 0.001); males were more likely to answer "Not true" and less likely to answer "Somewhat True" and "Certainly True" (all post-hoc residuals $\geq \pm 2.983$, ps $\leq 0.017$) compared to females. Peer problems

responses were mostly similar across both sexes, although the item "I have one good friend or more" was different between sex ($\chi^2$ = 10.970, df = 2, p = 0.004), with males more likely to answer "Somewhat True" (post-hoc residual = 2.877, p = 0.024) and less likely to answer "Certainly True" (post-hoc residual = -3.290, p = 0.006) compared to females. Most parents responded "No/Does not apply" to socioeconomic stress items and the distribution was similar between sexes (all $\chi^2 \leq 4.459$, df = 2, ps $\geq$ 0.108).

## Measurement invariance

Strict measurement invariance was achieved for parent-reported socioeconomic stress and family support, as well as child-reported peer problems and emotional symptoms. This showed that the same construct was being measured between sex and it allowed comparison of latent mean values between sex. Full results for the measurement invariance analysis are presented in S2 Appendix, and S1 and S2 Tables in S1 File. There was no significant difference in the latent mean values between sex for socioeconomic stress (estimate = 0.040, SE = 0.075, p = 0.595) or family support (estimate = -0.083, SE = 0.066, p = 0.205). The mean value for males was larger for peer problems (estimate = 0.136, SE = 0.065, p = 0.036) and smaller for emotional symptoms compared to females (estimate = -0.926, SE = 0.075, p < 0.001). There were some items with low standardised loadings (< 0.50) for both sexes in the measurement invariance models–'problems with neighbours/neighbourhood' for socioeconomic stress and 'I get a lot of headaches, stomach-aches or sickness' for emotional symptoms. Fixing the loadings of these items to zero in a separate models resulted in significantly worse model fit (socioeconomic stress: $\Delta\chi2$ = 28.561, $\Delta$df = 1, p < 0.001; emotional symptoms: ($\Delta\chi2$ = 216.89, $\Delta$df = 1, p < 0.001), therefore these items were retained in the model. Additional information on the potential impact of the number of non-zero data points for the socioeconomic stress latent variable is described in S2 Appendix in S1 File.

## Structural equation modelling

First, model 1 was assessed independently and this was an adequate fit to the data (robust $\chi^2$ = 986.381, p-value < 0.001, robust RMSEA = 0.26 [0.023, 0.029], robust CFI = 0.924). For both females and males, peer problems were a positive predictor of emotional symptoms (males β = 0.622, p < .001; females β = 0.495, p < .001), socioeconomic stress was a negative predictor of family support (males β = -0.187, p < .001; females β = -0.342, p < .001). Furthermore, there was evidence for sex-specific findings. For females, socioeconomic stress was a negative predictor of vmPFC GMV (β = -0.124, p = 0.008) and for males, socioeconomic stress was a negative predictor of emotional symptoms (β = -0.115, p = 0.046) and amygdala GMV (β = -0.098, p = 0.033). Furthermore, there was significant covariance between amygdala and vmPFC GMV (males β = 0.303, p < 0.001; females β = 0.333, p < 0.001). Other relationships of interest were not statistically significant.

Next, model 1 was nested within model 2, which resulted in a poor fit to the data (see Table 4). Model 2 was a comparatively better fit in terms of the chi-square difference test and improvement in CFI value. The CFI value was just below the standard criteria of 0.95 and the chi-square value was significant, indicating sub-optimal fit. However, the latter is common in models with large sample sizes [55].

**Table 4. Robust fit statistics for the nested models, including chi-square statistic, df, chi-square difference tests, CFI and RMSEA with 90% CI (N = 1950).**

| Model | $\chi^2$ | df | p | $\Delta\chi^2$ | $\Delta$df | p | CFI | RMSEA [90% CI] |
|---|---|---|---|---|---|---|---|---|
| 1 | 1925.773 | 656 | < .001 | - | - | - | 0.777 | 0.045 [0.042, 0.047] |
| 2 | 1015.382 | 626 | < .001 | 557.12 | 30 | < .001 | 0.932 | 0.025 [0.022, 0.028] |

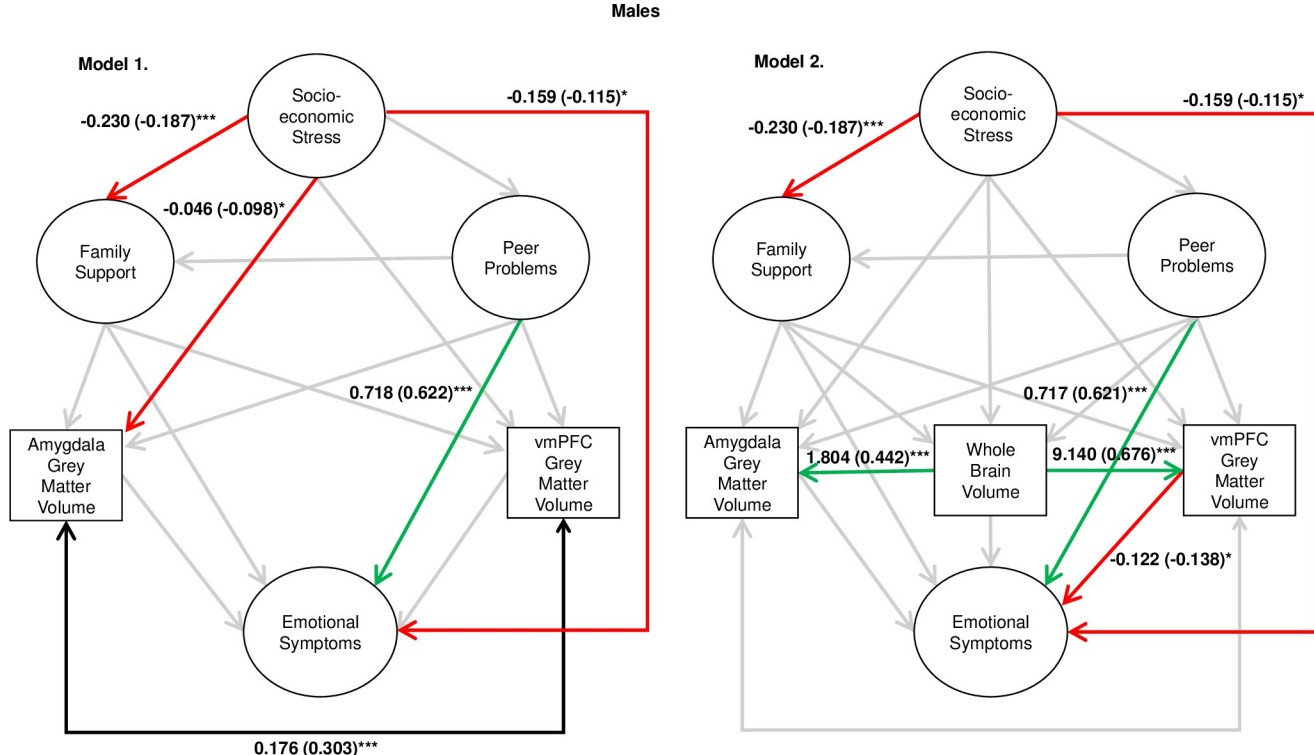

**Fig 3. Results for models 1 and 2 for the male sample.** Note. *** = p < .001, ** = p < .01, * = p < .05. Estimates are unstandardised path coefficients (standardised in parentheses). Amygdala and vmPFC GMV values were divided by 1,000, and WBV was divided by 1,000,000, so that the values were closer in magnitude to other variables in the model.

Statistics for the associations of interest for models 1 and 2 are depicted in Fig 3 for males, and Fig 4 for females. For the full regression statistics, see S3 Table for model 1 and S4 Table for model 2 in S1 File. In model 2, the associations between peer problems and emotional symptoms, and socioeconomic stress and family support, remained statistically significant for both males and females. Socioeconomic stress was again found to be a negative predictor of emotional symptoms in males only. However, the sex-specific associations between socioeconomic stress and amygdala/vmPFC GMV were non-significant in this model. Instead, after accounting for the strong association between WBV and regional GMV, for males vmPFC GMV was a negative predictor of emotional symptoms ($\beta$ = -0.138, p = 0.022) and, for females, socioeconomic stress was found to negatively predict WBV ($\beta$ = -0.127, p = 0.007). In all models, peer problems were not a significant predictor of family support in neither males nor females, therefore a mediation analysis was not conducted.

**Testing sex differences.** In model 2, there was no significant difference in model fit when coefficients were constrained to equality by sex for peer problems as a predictor of emotional symptoms ($\chi^2$ = 2.284, df = 1, p = 0.131) and for socioeconomic stress as a predictor of family support ($\chi^2$ = 2.675, df = 1, p = 0.102) which suggests no sex differences in the magnitude of the relationships.

**Sensitivity analysis.** *Parental education.* To check the validity of the latent variable of socioeconomic stress, we investigated whether it was predicted by a more objective marker of socioeconomic status—parental education. The addition of parental education to the model also allowed us to test whether the significant associations found related to socioeconomic stress were explained by parental education.

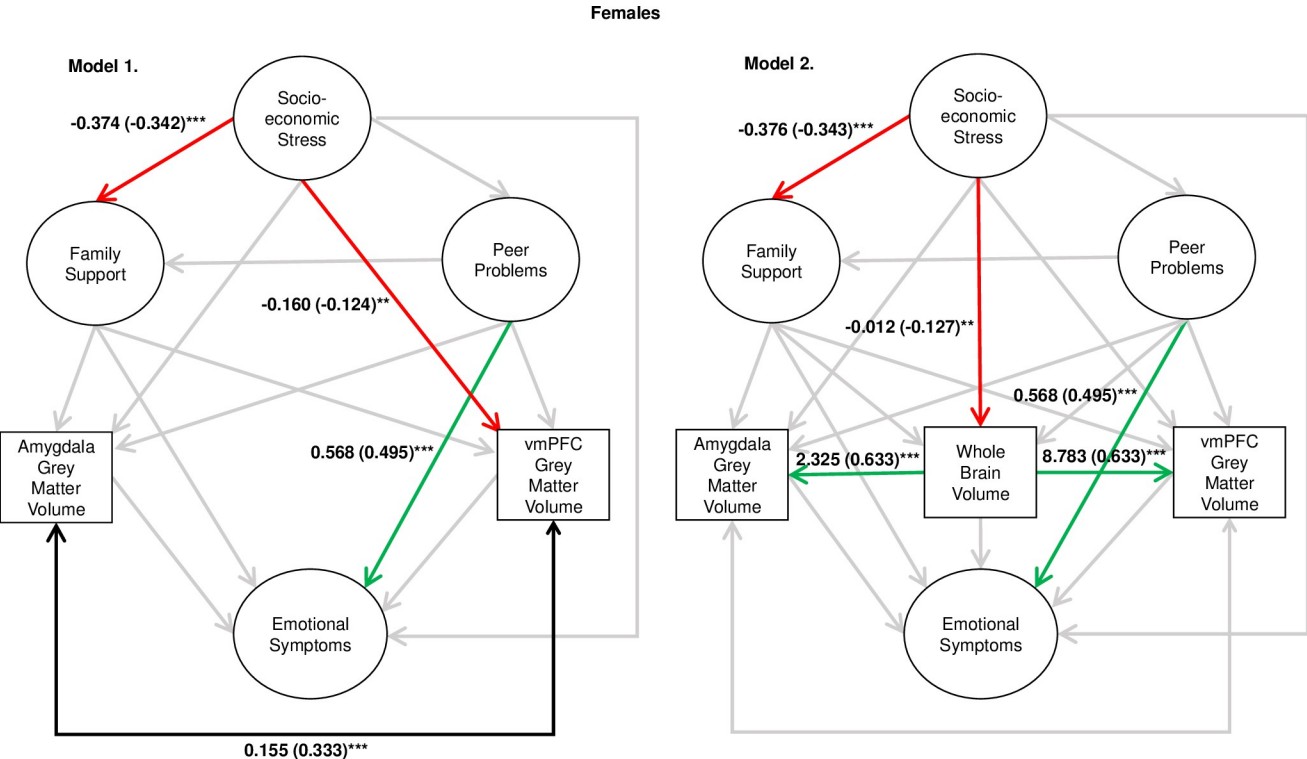

**Fig 4. Results for models 1 and 2 for the female sample.** Note. *** = p < .001, ** = p < .01, * = p < .05. Estimates are unstandardised path coefficients (standardised in parentheses). Amygdala and vmPFC GMV values were divided by 1,000, and WBV was divided by 1,000,000, so that the values were closer in magnitude to other variables in the model.

Parental education was added into model 2 as a predictor of: socioeconomic stress, emotional symptoms, family support, peer problems, WBV, amygdala GMV and vmPFC GMV. We hypothesised that parental education would be negatively associated with socioeconomic stress. We also predicted that the associations of interest would remain statistically significant as in model 2 with the addition of parental education.

Parental education was comprised of both mother's and father's highest education (8-point scale, 1 = Professional qualification e.g., PhD, MD, Master's, 8 = None) and the data were present for most participants in the sample (n = 1938). Values were reverse-scored and summed for both mother and father so that a higher score indicated higher combined educational achievement.

The model was a good fit to the data: robust $\chi^2$ = 1019.611, p-value < 0.001, robust CFI = 0.934, robust RMSEA = 0.024 [0.021, 0.027]. Regression results are found in S5 Table in S1 File.

As predicted, higher parental education was associated with lower socioeconomic stress (male/female β = -0.250/-0.241, p < 0.001), which provides evidence for the validity of socioeconomic stress.

The other main findings are as follows:

- Peer problems positively predicted emotional symptoms for males (β = 0.623, p < 0.001) and females (β = 0.494, p < 0.001). Parental education did not predict emotional symptoms for either sex.

- Socioeconomic stress negatively predicted family support for males (β = -0.177, p = 0.001) and females (β = -0.314, p < 0.001). Parental education positively predicted family support for females only (β = 0.118, p = 0.010).

- For females, socioeconomic stress negatively predicted whole brain volume ($\beta$ = -0.105, p = 0.027). Parental education positively predicted whole brain volume for both males ($\beta$ = 0.148, p < 0.001) and females ($\beta$ = 0.109, p = 0.002).

- For males, vmPFC GMV negatively predicted emotional symptoms ($\beta$ = -0.139, p = 0.019). Parental education did not predict vmPFC GMV.

- However, for males, socioeconomic stress no longer significantly predicted emotional symptoms ($\beta$ = -0.105, p = 0.071).

The findings remained largely the same, which suggests that these effects are not due to the confounding effects of parental education. The only significant difference in results is that socioeconomic stress was no longer a statistically significant negative predictor of emotional symptoms for males.

*Psychiatric diagnosis.* Psychiatric diagnosis was included as a covariate in the study, but sex biases in the frequencies of psychiatric disorders may have influenced the findings. The distribution of psychiatric diagnoses by sex are presented in S6 Table in S1 File. There were more males with an ADHD/Autism diagnosis than females, and more females with a mood or anxiety disorder compared to males. Information on main diagnosis was not available, so investigating the effect of dummy-coded diagnoses in the same model resulted in model non-convergence due to multi-collinearity of comorbid diagnoses. Instead, we ran two additional models: one that excluded participants with any psychiatric diagnosis (see S7 Table in S1 File for regression output) and one that only investigated mood or anxiety disorder diagnosis instead of any psychiatric diagnosis (see S8 Table in S1 File), due to their high likelihood of comorbidity and given the focus on emotional symptoms in the current study.

Both models showed good fit to the data. For the psychiatric diagnosis excluded model in S7 Table in S1 File, there were zero responses for males for the "Not True" option for the "Gets love and affection" item in the Family Life Questionnaire, therefore the responses to "Not True" and "Somewhat True" were merged in this model. In both models, main associations of interest found in previous models remained statistically significant. Additionally, family support was negatively associated with emotional symptoms in females only in both models.

## Discussion

The current study aimed to explore a multidisciplinary perspective of the influence of social factors and brain structure on emotional symptoms in early adolescence. The results indicated that, for both males and females, peer problems were positively associated with emotional symptoms, and socioeconomic stress was negatively associated with family support at age 14 years. Additionally, sex differences were observed: for males, vmPFC GMV was negatively associated with emotional symptoms, and, for females, socioeconomic stress was negatively associated with WBV. However, socioeconomic stress and family support were not associated with regional brain structure or emotional symptoms. Family support was negatively associated with emotional symptoms in females only in the sensitivity analysis, where models either did not include participants with a psychiatric diagnosis or only included participants with mood or anxiety disorders. Peer problems were not a significant predictor of family support: family support did not mediate the relationship between peer problems and emotional symptoms.

Peer problems were a positive predictor of emotional symptoms. Subsequent analyses found that the strength of this relationship was similar for both males and females (see Results sub-section 'Testing sex differences'), which underscores the importance of peer relationships for mental health at this age for both sexes. The finding, in line with previous research, showed

peer exclusion and victimisation have a deleterious impact on adolescent mental health [23, 24]. Furthermore, this finding is related to the notion that good peer relationships are important in adolescence, and any threats to them affect mental health [3, 22]. Previous research suggested that female mental health may be more affected by relational victimisation than male mental health [12, 26, 27], although the current study found that peer problems have a similar negative effect on both male and female adolescent mental health. This may be due to different conceptualisations of relational victimisation and peer problems. Bullying and victimisation was only one component of the latent variable of 'peer problems' in the current study; additional components included preference for being alone, having one good friend or more, etc. Therefore, peer problems were more broadly defined, and reflected issues with exclusion or disconnection along with victimisation. However, measurement invariance tests confirmed conceptual equivalence of peer problems between sexes, so this supports the idea that peer problems at its core affects mental health similarly for males and females at this age.

In addition, socioeconomic stress was a negative predictor of family support, even when parental education was factored into the model. This supports the Family Stress Model, which posits that socioeconomic difficulties result in decreased parental availability and support for their children [4, 33–36]. Initially in the WBV-included model, socioeconomic stress was a negative predictor of emotional symptoms in males, however this finding was non-significant when parental education was added into the model as part of the sensitivity analysis. Therefore, the relationship could be partly explained by parental education, which reflects parental status or resources. Interestingly, parental education was significantly positively associated with family support for females only. This suggests that parental education may be associated with support specific to gender-differentiated parenting practices. A meta-analysis found that parents used more autonomy-supportive strategies–which includes affirmation as used in the current study–towards girls rather than boys when looking at studies from the 1990s onwards. Before the 1990s, the effect was found in boys instead, which reflects cultural changes in parenting practices, and shows how notions of support are dependent on cultural norms [66]. Socioeconomic stress and parental education were not directly associated with emotional symptoms for males and females. This was unexpected given the wealth of research linking low socioeconomic status with poor adolescent mental health for both males and females [31, 32]. Because the current study uses cross-sectional data, we are unable to determine the temporality of socioeconomic factors and family support, and possible sex differences. Future longitudinal analyses will be able to untangle these relationships and whether there is an effect on adolescent emotional symptoms.

Smaller vmPFC GMV, after correcting for WBV, was associated with greater emotional symptoms for males only at this age. Previous cross-sectional analyses found that onset of adolescent depression was associated with reduced volume of frontal regions, including the orbitofrontal cortex, which was used as the definition of the vmPFC in the current study [67]. For males, the amygdala and vmPFC has a delayed maturational path compared to females [8, 9]. A smaller vmPFC volume may reflect maturational delays compared to other males, which reflects an attenuated ability for frontal regions to downregulate subcortical regions, leading to increased emotional distress. However, due to the cross-sectional nature of our study, the maturational pattern of regions cannot be established, and those conclusions are tentative. This finding reveals the impact of absolute regional differences for male adolescents, but that does not tell us whether the region has matured or is still maturing for a particular person.

Another sex-specific finding was that socioeconomic stress was negatively associated with WBV in females only. Previous research has indicated that objective measures of socioeconomic status, such as family income, occupation, and education, are associated with WBV and total brain surface area [68–70]. The current study also found that parental education

predicted WBV in both males and females, but this study goes further to show that stress from socioeconomic conditions affect whole brain volume, which is in line with studies reporting the deleterious effect of stress on the developing brain [71], and that has a stronger effect on females, which may be due to their increased sensitivity to stress compared to males [35]. Further research should clarify whether socioeconomic stress has a distributed effect on the female brain, or whether specific regions are impacted, and whether this affects other cognitive or emotional symptoms.

Family support did not directly influence emotional symptoms in models that controlled for any psychiatric diagnosis, nor did it mediate the effect of peer problems on emotional symptoms in any model. In the sensitivity analysis, models that either did not include participants with a psychiatric diagnosis or only included participants with mood or anxiety disorders found that family support was negatively associated with emotional symptoms in females only. This suggests that the link between family support and emotional symptoms in females was previously obscured by the inclusion of participants who had psychiatric diagnoses other than mood or anxiety disorders. These findings contradict previous research that found that, similarly for both sexes, family support independently predicts mental health outcomes [23] and buffers against the effect of peer problems on mental health [28, 29]. Females may be more sensitive to general family support, or it may be that the type of support needs to be targeted to the problem for it to have an effect. Successful social support has been found to depend on the source, type, and timing of the support [72], suggesting that general measures of family support may not be sensitive to determine a buffering effect for both sexes. In addition, previous studies measured adolescent perceptions of family support rather than parent perceptions as was the case in the current study. Parent reports may be biased because they may only report positive characteristics due to social desirability. This is congruent with the data, as many of the family support items were positively affirmed by the majority of parents.

We found no association between amygdala and vmPFC GMV, and family support, peer problems, or socioeconomic stress in the best-fitting model. The amygdala and vmPFC were chosen as regions of interest due to their involvement in emotional regulation [44] and their distinct maturational profiles across adolescence [1]. The measures used or the design employed in the current study may not be able to uncover the effect of the social environment on the developing brain. The current data only provides a snapshot of the peer and family dynamics within an adolescent's life; investigating changes over time may be more fitting to the protracted process of brain development. In addition, the current study highlights the importance of WBV correction when investigating regional brain differences. Socioeconomic stress was associated with amygdala GMV in males and vmPFC GMV in females when uncorrected for WBV, however this association was attenuated and not statistically significant when WBV was included as a covariate and a predictor in separate models.

## Strengths and limitations

One of the strengths of the current study is the use of a model with a multidisciplinary perspective that was tested in a large dataset. That allowed the investigation of three frames of reference: socioeconomic conditions, social relationships, and brain structure, providing an integrated view of adolescent mental health [5]. The large sample size ensured that the study had the statistical power to detect robust effects that are less likely to be spurious [73]. Another strength is the use of analytic techniques such as measurement invariance and SEM. Establishing measurement invariance allowed us to formally specify that the same latent variables were measured between sex and that differences are not simply due to measurement error [74]. SEM which allows simultaneously modelling of complex relationships between variables [75].

Considering factors in isolation may lead to a significant result, but this may be influenced by interactions with other factors when included in the model. Therefore, simultaneous modelling allowed us to determine the relative strength of effects in the presence of other variables, strengthening the validity of the results.

Limitations of the study include the lack of child-reported measures for family support. Perceptions of support are strongly associated with mental health outcomes, even if there is a weak association with objective indicators of support [76]. Therefore, parents may report supportive behaviours, but it may not be perceived as supportive or helpful to the adolescent. Indeed, other measures of parent and child reports of family support have found discrepancies. Correlations between parent and child reports of parent support are weak [77], with parents reporting themselves to be more supportive compared to child reports [78]. Importantly, adolescents who reported poorer parent practices compared to parents were at higher risk of internalising symptoms [78]; this discrepancy therefore reveals information about the adult-child relationship that has implications for mental health. Unfortunately, the Family Life Questionnaire in the current study is parent-reported only, and other measures of child-reported family support were not available in the IMAGEN dataset, so this could not be explored in the current study. Future studies should aim to assess discrepancies between parent and child reports of family support in different datasets.

IMAGEN is a multi-centre study designed to maximise sample size. Different scanners are used at different sites for the neuroimaging assessment. To minimise variability between sites, a central protocol was used between sites and quality control and pre-processing procedures were implemented, explained in depth elsewhere [38]. Recruitment centre was included as a covariate in the analysis to further account for potential homogeneity. However, it is acknowledged that variability between sites could have affected the results in the current analysis.

The use of cross-sectional data is also limiting because we were unable to investigate developmental trajectories over time. There are significant individual differences in brain development in terms of the intercept and slope of change over time [2]. Environmental variables have been shown to affect the maturation of the brain across adolescence, such as parental support [6] and socioeconomic factors [7, 37]. Therefore, future research should look at brain development longitudinally, to detect individual differences in the developmental trajectory of the brain, the impact of environmental variables, and how this relates to emotional functioning.

Sex differences were investigated in the current study, however we were unable to investigate the role of gender non-conformity due to this information not being available. Gender non-conformity could have influenced the study findings, due to effects on depressive symptoms and bullying victimisation [79]. Future studies could look at both sex and gender differences in the role of social and neurobiological factors in emotional symptoms.

## Implications

Peer problems influenced emotional symptoms in early adolescence, highlighting the need to promote social integration for good mental health. Schools can play a critical role in this, using programs to promote supportive peer relationships and to focus on social skill development [80].

Additionally, socioeconomic stress was found to have a downstream effect on both family support for both sexes and WBV for females. This reveals the complex, often subtle relationships between variables and suggests that socioeconomic stress may be a target for intervention. Objective indicators of socioeconomic status such as parental occupation, income or education are difficult and timely to modify, however interventions to help with managing

stress in relation to socioeconomic circumstances may be an achievable step in improving the family environment and resulting biological impact. For example, the Family Adjustment and Adaptation Response Model [81] posits that family stress can be managed by using resources from multiple levels–individuals, family and community–to meet demands that are leading to stress. In this way, a multiple level approach can be used to deal with a multiple level problem. Cognitive-based interventions have demonstrable effectiveness in managing adolescent stress [82], therefore even in families with high socioeconomic stress and low support, there are person-centred avenues that can help protect adolescent mental health.

## Significance statement

Using structural equation modelling in a large dataset (IMAGEN), we investigated the nuanced associations between socioeconomic conditions, social relationships, and regional brain structure in predicting adolescent emotional symptoms, separately for each sex. Using this approach, we found significant associations that were common to both sexes, and associations that were sex specific. Future research should aim to verify the associations using longitudinal data, to assess the directionality of relationships of how both social and biological factors affect mental health in adolescence.

## Conclusions

At age 14 years, problems with peers were significantly associated with emotional symptoms for both males and females. Family socioeconomic stress was related to family support and female brain volume. Future longitudinal study should assess how socioeconomic conditions, social relationships, and brain structure interact prospectively to affect mental health.

## Supporting information

**S1 File. Its contains all supporting appendices and tables.**
(DOCX)

## Acknowledgments

IMAGEN Consortium author list:

Arun L.W. Bokde[7], Sylvane Desrivières[8], Herta Flor[5,9], Antoine Grigis[10], Hugh Garavan[11], Penny Gowland[12], Andreas Heinz[13], Rüdiger Brühl[14], Jean-Luc Martinot[15], Marie-Laure Paillère Martinot[16], Eric Artiges[17], Dimitri Papadopoulos Orfanos[10], Tomáš Paus [18, 19], Luise Poustka[20], Sarah Hohmann[4], Sabina Millenet[4], Juliane H. Fröhner[21], Lauren Robinson[22], Michael N. Smolka[21], Henrik Walter[13], Jeanne Winterer[13, 23], Robert Whelan[24], Gunter Schumann[4, 25]

[7] Discipline of Psychiatry, School of Medicine and Trinity College Institute of Neuroscience, Trinity College Dublin, Dublin, Ireland; [8] Centre for Population Neuroscience and Precision Medicine (PONS), Institute of Psychiatry, Psychology & Neuroscience, SGDP Centre, King's College London, United Kingdom; [9] Department of Psychology, School of Social Sciences, University of Mannheim, 68131 Mannheim, Germany; [10] NeuroSpin, CEA, Université Paris-Saclay, F-91191 Gif-sur-Yvette, France; [11] Departments of Psychiatry and Psychology, University of Vermont, 05405 Burlington, Vermont, USA; [12] Sir Peter Mansfield Imaging Centre School of Physics and Astronomy, University of Nottingham, University Park, Nottingham, United Kingdom; [13] Department of Psychiatry and Psychotherapy CCM, Charité – Universitätsmedizin Berlin, corporate member of Freie Universität Berlin, Humboldt-Universität zu Berlin, and Berlin Institute of Health, Berlin, Germany; [14] Physikalisch-Technische

Bundesanstalt (PTB), Braunschweig and Berlin, Germany; [15] Institut National de la Santé et de la Recherche Médicale, INSERM U1299 "Developmental trajectories & psychiatry""; Université Paris-Saclay, Ecole Normale supérieure Paris-Saclay, CNRS, Centre Borelli; Gif-sur-Yvette, France; [16] Institut National de la Santé et de la Recherche Médicale, INSERM U1299 "Developmental trajectories & psychiatry""; Université Paris-Saclay, Ecole Normale supérieure Paris-Saclay, CNRS, Centre Borelli; Gif-sur-Yvette; and AP-HP.Sorbonne Université, Department of Child and Adolescent Psychiatry, Pitié-Salpêtrière Hospital, Paris, France; [17] Institut National de la Santé et de la Recherche Médicale, INSERM U1299 "Developmental trajectories & psychiatry""; Université Paris-Saclay, Ecole Normale supérieure Paris-Saclay, CNRS, Centre Borelli; Gif-sur-Yvette; and Etablissement Public de Santé (EPS) Barthélemy Durand, 91700 Sainte-Geneviève-des-Bois, France; [18] Departments of Psychiatry and Neuroscience and Centre Hospitalier Universitaire Sainte-Justine, University of Montreal, Montreal, Quebec, Canada; [19] Departments of Psychiatry and Psychology, University of Toronto, Toronto, Ontario, Canada; [20] Department of Child and Adolescent Psychiatry and Psychotherapy, University Medical Centre Göttingen, von-Siebold-Str. 5, 37075, Göttingen, Germany; [21] Department of Psychiatry and Neuroimaging Center, Technische Universität Dresden, Dresden, Germany; [22] Department of Psychological Medicine, Section for Eating Disorders, Institute of Psychiatry, Psychology and Neuroscience, King's College London, London, SE5 8AF, UK; [23] Department of Education and Psychology, Freie Universität Berlin, Berlin, Germany; [24] School of Psychology and Global Brain Health Institute, Trinity College Dublin, Ireland; [25] PONS Research Group, Dept of Psychiatry and Psychotherapy, Campus Charite Mitte, Humboldt University, Berlin and Leibniz Institute for Neurobiology, Magdeburg, Germany, and Institute for Science and Technology of Brain-inspired Intelligence (ISTBI), Fudan University, Shanghai, P.R. China.

## Author Contributions

**Conceptualization:** Jessica Stepanous, Luke Munford, Rebecca Elliott.

**Formal analysis:** Jessica Stepanous.

**Methodology:** Jessica Stepanous, Luke Munford, Pamela Qualter.

**Project administration:** Jessica Stepanous.

**Resources:** Tobias Banaschewski, Frauke Nees.

**Supervision:** Luke Munford, Pamela Qualter, Tobias Banaschewski, Frauke Nees, Rebecca Elliott.

**Visualization:** Jessica Stepanous.

**Writing – original draft:** Jessica Stepanous.

**Writing – review & editing:** Luke Munford, Pamela Qualter, Frauke Nees, Rebecca Elliott.

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
