## [Decision Letter · Decision Letter 0]

1 Jun 2022

PONE-D-21-30637Social environment and brain structure in adolescent mental health: A cross-sectional structural equation modelling study using IMAGEN dataPLOS ONE

Dear Dr. Stepanous,

Thank you for submitting your manuscript to PLOS ONE. After careful consideration, we feel that it has merit but does not fully meet PLOS ONE’s publication criteria as it currently stands. Both reviewers have raised some questions that will need to be addressed. Therefore, we invite you to submit a revised version of the manuscript that addresses the points raised during the review process. 

We look forward to receiving your revised manuscript.

Kind regards,

Therese van Amelsvoort

Academic Editor

PLOS ONE

Journal Requirements:

“This work received support from the following sources: the ESRC-BBSRC Soc-B Centre for Doctoral Training (ES/P000347/1), the European Union-funded FP6 Integrated Project IMAGEN (Reinforcement-related behaviour in normal brain function and psychopathology) (LSHM-CT- 2007-037286), the Horizon 2020 funded ERC Advanced Grant ‘STRATIFY’ (Brain network based stratification of reinforcement-related disorders) (695313), Human Brain Project (HBP SGA 2, 785907, and HBP SGA 3, 945539), the Medical Research Council Grant 'c-VEDA’ (Consortium on Vulnerability to Externalizing Disorders and Addictions) (MR/N000390/1), the National Institute of Health (NIH) (R01DA049238, A decentralized macro and micro gene-by-environment interaction analysis of substance use behavior and its brain biomarkers), the National Institute for Health Research (NIHR) Biomedical Research Centre at South London and Maudsley NHS Foundation Trust and King’s College London, the Bundesministeriumfür Bildung und Forschung (BMBF grants 01GS08152; 01EV0711; Forschungsnetz AERIAL 01EE1406A, 01EE1406B; Forschungsnetz IMAC-Mind 01GL1745B), the Deutsche Forschungsgemeinschaft (DFG grants SM 80/7-2, SFB 940, TRR 265, NE 1383/14-1), the Medical Research Foundation and Medical Research Council (grants MR/R00465X/1 and MR/S020306/1), the National Institutes of Health (NIH) funded ENIGMA (grants 5U54EB020403-05 and 1R56AG058854-01).  Further support was provided by grants from: - the ANR (ANR-12-SAMA-0004, AAPG2019 - GeBra), the Eranet Neuron (AF12-NEUR0008-01 - WM2NA; and ANR-18-NEUR00002-01 - ADORe), the Fondation de France (00081242), the Fondation pour la Recherche Médicale (DPA20140629802), the Mission Interministérielle de Lutte-contre-les-Drogues-et-les-Conduites-Addictives (MILDECA), the Assistance-Publique-Hôpitaux-de-Paris and INSERM (interface grant), Paris Sud University IDEX 2012, the Fondation de l’Avenir (grant AP-RM-17-013 ), the Fédération pour la Recherche sur le Cerveau; the National Institutes of Health, Science Foundation Ireland (16/ERCD/3797), U.S.A. (Axon, Testosterone and Mental Health during Adolescence; RO1 MH085772-01A1), and by NIH Consortium grant U54 EB020403, supported by a cross-NIH alliance that funds Big Data to Knowledge Centres of Excellence, the National Institute for Health Research (NIHR) Biomedical Research Centre (BRC) at South London and Maudsley NHS Foundation Trust (SLaM)  and King’s College London (KCL). “

“JS is funded by the ESRC-BBSRC Soc-B Centre for Doctoral Training (ES/P000347/1). The IMAGEN study was funded by the European Union-funded FP6 Integrated Project IMAGEN (Reinforcement-related behaviour in normal brain function and psychopathology) (LSHM-CT- 2007-037286), the Horizon 2020 funded ERC Advanced Grant ‘STRATIFY’ (Brain network based stratification of reinforcement-related disorders) (695313), Human Brain Project (HBP SGA 2, 785907, and HBP SGA 3, 945539), the Medical Research Council Grant 'c-VEDA’ (Consortium on Vulnerability to Externalizing Disorders and Addictions) (MR/N000390/1), the National Institute of Health (NIH) (R01DA049238, A decentralized macro and micro gene-by-environment interaction analysis of substance use behavior and its brain biomarkers), the National Institute for Health Research (NIHR) Biomedical Research Centre at South London and Maudsley NHS Foundation Trust and King’s College London, the Bundesministeriumfür Bildung und Forschung (BMBF grants 01GS08152; 01EV0711; Forschungsnetz AERIAL 01EE1406A, 01EE1406B; Forschungsnetz IMAC-Mind 01GL1745B), the Deutsche Forschungsgemeinschaft (DFG grants SM 80/7-2, SFB 940, TRR 265, NE 1383/14-1), the Medical Research Foundation and Medical Research Council (grants MR/R00465X/1 and MR/S020306/1), the National Institutes of Health (NIH) funded ENIGMA (grants 5U54EB020403-05 and 1R56AG058854-01). Further support was provided by grants from: - the ANR (ANR-12-SAMA-0004, AAPG2019 - GeBra), the Eranet Neuron (AF12-NEUR0008-01 - WM2NA; and ANR-18-NEUR00002-01 - ADORe), the Fondation de France (00081242), the Fondation pour la Recherche Médicale (DPA20140629802), the Mission Interministérielle de Lutte-contre-les-Drogues-et-les-Conduites-Addictives (MILDECA), the Assistance-Publique-Hôpitaux-de-Paris and INSERM (interface grant), Paris Sud University IDEX 2012, the Fondation de l’Avenir (grant AP-RM-17-013 ), the Fédération pour la Recherche sur le Cerveau; the National Institutes of Health, Science Foundation Ireland (16/ERCD/3797), U.S.A. (Axon, Testosterone and Mental Health during Adolescence; RO1 MH085772-01A1), and by NIH Consortium grant U54 EB020403, supported by a cross-NIH alliance that funds Big Data to Knowledge Centres of Excellence, the National Institute for Health Research (NIHR) Biomedical Research Centre (BRC) at South London and Maudsley NHS Foundation Trust (SLaM) and King’s College London (KCL). The funders had no role in study design, data collection and analysis, decision to publish, or preparation of the manuscript.”

3. Thank you for stating the following in the Competing Interests/Financial Disclosure* (delete as necessary) section:

“Dr Banaschewski served in an advisory or consultancy role for Lundbeck, Medice, Neurim Pharmaceuticals, Oberberg GmbH, Shire. He received conference support or speaker’s fee by Lilly, Medice, Novartis and Shire. He has been involved in clinical trials conducted by Shire & Viforpharma. He received royalties from Hogrefe, Kohlhammer, CIP Medien, Oxford University Press. The present work is unrelated to the above grants and relationships. The other authors report no biomedical financial interests or potential conflicts of interest.”

We note that one or more of the authors are employed by a commercial company

4. One of the noted authors is a group or consortium [IMAGEN Consortium  ]. In addition to naming the author group, please list the individual authors and affiliations within this group in the acknowledgments section of your manuscript. Please also indicate clearly a lead author for this group along with a contact email address.

Reviewers' comments:

Reviewer's Responses to Questions

**Comments to the Author**

1. Is the manuscript technically sound, and do the data support the conclusions?

Reviewer #1: Yes

Reviewer #2: Yes

2. Has the statistical analysis been performed appropriately and rigorously? 

Reviewer #1: Yes

Reviewer #2: Yes

3. Have the authors made all data underlying the findings in their manuscript fully available?

Reviewer #1: Yes

Reviewer #2: No

4. Is the manuscript presented in an intelligible fashion and written in standard English?

Reviewer #1: Yes

Reviewer #2: Yes

5. Review Comments to the Author

Reviewer #1: The study aims first in identifying whether emotional symptoms are predicted by the concurrent effects of social factors (socioeconomic stress, family support, peer problems) and brain structure (amygdala and ventromedial prefrontal cortex volume) in 14 year old adolescents. Second, whether family support buffers against any negative effects of peer problems on mental health and finally whether social factors affect brain structure resulting in emotional symptoms.

Males and females are treated as separate groups. Volumetric analysis was done with Freesurfer on data from 8 different centres. Social factors were rated based on 3 or 4 point scales. Associations were established using multi-group structural equation modelling (SEM) in 2 models (with and without whole brain volume correction). The results indicate that for both sexes, peer problems were positively associated with emotional symptoms and socioeconomic stress was negatively associated with family support. Furthermore, socioeconomic stress and ventromedial prefrontal cortex grey matter volume was negatively associated with emotional symptoms for males when corrected for whole brain volume, and socioeconomic stress was negatively associated with whole brain volume for females. Family support was not found mediating the relationship between peer problems and emotional symptoms.

The abstract doesn’t mention the type of association of the sex specific findings in the full models: [‘socioeconomic stress and ventromedial prefrontal cortex grey matter volume was associated with emotional symptoms for males when corrected for whole brain volume, and socioeconomic stress was associated with whole brain volume for females‘].

The introduction clearly explains why these social factors and brain structures were chosen and whether their impact is different on sexes. What is not so clear is why these levels interact to affect mental health risk and resilience. Can the authors add more references regarding developmental mismatch and further evidence on adolescent depression being associated with reductions in frontal regions and a trend towards a smaller amygdala [reference 19 only indicates a trend with no statistical significance]?

In the methods it is stated that one of the predictors of emotional symptoms is ‘perceived sex’. Neither SEM model is shown containing this predictor (Figure 2). Details of the MRI protocol should be included. At the very least describing the T1 acquisition. The version of FreeSurfer is not mentioned. The supplementary tables are a very useful addition.

For the results, Table S1 shows all the configural family support models having large p-values, but the fit is perfect. Can the authors explain that? There is a further analysis (strength of peer problem-emotional symptoms relationship) mentioned in the 2nd paragraph of the discussion, but this is not shown. This could be included in the supplementary material.

The discussion needs more support for the Family Stress model. Also, more recent references might demonstrate findings similar to this study and regarding SES and mental health being ‘irrespective of gender’. It is stated that ‘This study goes further to show that stress from socioeconomic conditions affect whole brain structure’. This may be misleading. As only the total brain volume was used (minus the ventricles, CSF and dura) and brain subdivisions were not considered individually, I suggest to rephrase this to ‘ whole brain volume’. instead of implying that the ‘whole structure is different. Which model is implied in ‘In this model, family support did not directly influence emotional symptoms, nor did it mediate the effect of peer problems on emotional symptoms’? The WBV correction is emphasised in the next paragraph, so I think that clarifying the model is important. Even though the authors have included the different centres as a covariate, I think that another limitation is lack of standardisation of the data. If I am not mistaken, the scanners and coils used were different and there is no discussion on how that affects the homogeneity of the dataset, regardless of implementing the same protocol everywhere (I assume this is the case!).

Typo: page 4: ‘This is important given retrospective reports show that half of all individuals’

Typo: S3: In the note.

Reviewer #2: The authors tested the hypothesis that sex determines/influences the relationships between social environment, brain structure (from MR), and emotional symptoms indicative for mental health issues. They did so in a large dataset (>2000) consisting of 14-year-olds because of the particular sensitivity to the social environment, and intense brain development at that age. The study concept is interesting and the study is mostly well-executed, but I have a number of concerns:

1. Page 7: “a systematic review found conflicting results for sex differences between socioeconomic status and mental health difficulties” � clarify sentence; it reads as though some words are missing.

2. Page 8: Incorporate research questions into the paragraph before (formulated in normal sentence form, not ?), instead of having a separate paragraph. What were your hypotheses?

3. Page 9, Participants: Where was ethics approval obtained?

4. Page 9, Measures: How did self-identified sex correspond to self-identified gender? Gender non-conformity may have a huge impact on the studied relationships with social, brain, and emotional variables. Please comment on this in the Discussion at the very least.

5. Page 9, Measures: For the questionnaires, what is the actual variable included in the SEM? Total (sum) score? Please describe.

6. Page 11, Family support: The family support is asked from the parental perspective, which may not correspond well with the child’s perspective (as alluded to in the Discussion). Are there data available on how well they correspond? Please report them. Perceived family support from the child’s perspective may be more predictive of mental health.

7. Page 11, Peer problems: Specify what you mean exactly rather than saying “response format is the same” (as something explained later on).

8. Page 12, Regional grey matter volume: Which atlas was used to extract the measures? Please specify.

9. Page 13, Outliers: How many outliers were there for each variable, and how many were multivariate outliers? Does the latter suggest potential for bias?

10. Page 13, Covariates: As I understood it, individuals received a “Yes” for the variable psychiatric diagnosis if they had any (one or more) psychiatric diagnosis, without regard for the type. This seems like a very general variable; it makes more sense to me to create a variable like this per main diagnosis, given the potential differentiated effects of various diagnoses, and the sex bias of certain diagnoses.

11. Page 13, Covariates: “Mean PDS scores were derived for males and females separately.” -> If the mean is calculated across an individual’s PDS items, it is self-evident that they are derived separately for males and females as well. Furthermore, is it customary to calculate the mean PDS score over the total PDS score?

12. Page 14, Analysis strategy: Instead of “These cases”, do you mean “96 cases”?

13. Page 14, Analysis strategy: “Next, measurement invariance [analysis] and …”

14. Page 15, Measurement invariance: the last paragraph about the numbers of response categories is not fully clear to me. Please clarify.

15. Page 19, Descriptive statistics: Which ‘spread’ statistic are you referring to specifically?

16. Page 20, Table 2: Also report t-statistics and df, not just p-vals.

17. Page 21: What psychiatric diagnoses were present in the cohort, how frequently, and how did their frequencies differ among the sexes? Is there confounding possible as a result?

18. Page 21: Most parents responded “No/NA” to socioeconomic stress items. What does this mean for your further analyses? Do you have enough non-zero socioeconomic stress data points to be able to reliably test the effect?

19. Page 21: Report t-test/chi-square test statistics for each of these group comparison statements. (t/X2, df, p)

20. Page 22-23: Report means/SD by sex (plus difference test) for questionnaire sum scores if that is what was included in the SEMs.

21. Page 24, Measurement invariance: Describe the main take home message from these analyses here, in addition to referring to the supplement.

22. Page 25: “Socioeconomic stress was again found to be a negative predictor of emotional symptoms in males only.” -> Could this unexpected direction of effect be because the stress measure is not sensitive enough? (see related comment above)

6. PLOS authors have the option to publish the peer review history of their article (what does this mean?). If published, this will include your full peer review and any attached files.

Reviewer #1: **Yes: **NEM van Haren

Reviewer #2: No

---

## [Author Response · Author response to Decision Letter 0]

19 Jul 2022

Dear Academic Editor and Reviewers,

Thank you for your comments in response to the submission to PLOS ONE entitled “Social environment and brain structure in adolescent mental health: A cross-sectional structural equation modelling study using IMAGEN data”.

Please find our response to your comments below.

Journal Requirements:

We can confirm that the manuscript meets PLOS ONE’s style requirements.

2. We note that you have provided additional information within the Acknowledgements Section that is not currently declared in your Funding Statement. Please note that funding information should not appear in the Acknowledgments section or other areas of your manuscript. We will only publish funding information present in the Funding Statement section of the online submission form. Please remove any funding-related text from the manuscript and let us know how you would like to update your Funding Statement.

Thank you for bringing this to our attention. We have updated the Acknowledgements Section and Funding Statement accordingly and presented it at the bottom of the cover letter.

3. Thank you for stating the following in the Competing Interests/Financial Disclosure* (delete as necessary) section...

We have updated the Funding Statement and Competing Interests Statement, and presented it at the bottom of the cover letter. The Author Contributions are the same as stated in the initial submission.

4. One of the noted authors is a group or consortium [IMAGEN Consortium ]. In addition to naming the author group, please list the individual authors and affiliations within this group in the acknowledgments section of your manuscript. Please also indicate clearly a lead author for this group along with a contact email address.

We have updated the Acknowledgements section with the authors and affiliations of the IMAGEN consortium. The manuscript has also been updated:

Line 8: the IMAGEN Consortium^

Lines 25-27: ‘^FN and TB are part of the IMAGEN Consortium. FN is the lead IMAGEN consortium author for this publication (nees@med-psych.unikiel.

de). The whole IMAGEN author list is provided in the Acknowledgements.’

Reviewer 1

The study aims first in identifying whether emotional symptoms are predicted by the concurrent effects of social factors (socioeconomic stress, family support, peer problems) and brain structure (amygdala and ventromedial prefrontal cortex volume) in 14 year old adolescents. Second, whether family support buffers against any negative effects of peer problems on mental health and finally whether social factors affect brain structure resulting in emotional symptoms.

Males and females are treated as separate groups. Volumetric analysis was done with Freesurfer on data from 8 different centres. Social factors were rated based on 3 or 4 point scales. Associations were established using multi-group structural equation modelling (SEM) in 2 models (with and without whole brain volume correction). The results indicate that for both sexes, peer problems were positively associated with emotional symptoms and socioeconomic stress was negatively associated with family support. Furthermore, socioeconomic stress and ventromedial prefrontal cortex grey matter volume was negatively associated with emotional symptoms for males when corrected for whole brain volume, and socioeconomic stress was negatively associated with whole brain volume for females. Family support was not found mediating the relationship between peer problems and emotional symptoms.

 Thank you for the helpful and detailed review of our manuscript.

The abstract doesn’t mention the type of association of the sex specific findings in the full models: [‘socioeconomic stress and ventromedial prefrontal cortex grey matter volume was associated with emotional symptoms for males when corrected for whole brain volume, and socioeconomic stress was associated with whole brain volume for females‘].

 Thank you for bringing this to our attention. We have amended this in the revised manuscript to explicitly state the direction of the association. This has also been amended in line with the sensitivity analysis of the inclusion of parental education following later reviewer comments – socioeconomic stress was no longer significantly associated with emotional symptoms. 

Lines 38-41: ‘ventromedial prefrontal cortex grey matter volume was negatively associated with emotional symptoms for males when corrected for whole brain volume, and socioeconomic stress was negatively associated with whole brain volume for females.’

The introduction clearly explains why these social factors and brain structures were chosen and whether their impact is different on sexes. What is not so clear is why these levels interact to affect mental health risk and resilience. Can the authors add more references regarding developmental mismatch and further evidence on adolescent depression being associated with reductions in frontal regions and a trend towards a smaller amygdala [reference 19 only indicates a trend with no statistical significance]?

 We have added additional references regarding developmental mismatch: 

‘The developmental mismatch hypothesis posits that subcortical regions mature faster than cortical regions [1,16,17]. This pattern of development has been used to explain the high emotional salience of peer relationships in adolescence and the resultant effect on social behaviour [17–19].” (lines 69-72). “These dramatic changes in the adolescent brain have the potential to explain adolescence as a sensitive period for onset of mental health difficulties [17–19].’ (lines 82-83).

We have re-worded the evidence that adolescent depression is associated with structural differences in frontal regions and the amygdala by the inclusion of a systematic review with inconsistent findings, which may be explained by the lack of consideration of sex differences: lines 83-89 “A systematic review looking at structural neuroimaging predictors of depression in childhood and adolescence found evidence for the role of reductions in prefrontal regions, however findings were not consistent. These inconsistencies were even more prevalent when looking other structures such as the amygdala [21]. One reason posited is due to a lack of consideration of sex differences in the studies. For example, one study found that onset of adolescent depression was associated with greater amygdala growth in females but attenuated growth in males between ages 12 and 16 years [6]. This reveals...” 

In the methods it is stated that one of the predictors of emotional symptoms is ‘perceived sex’. Neither SEM model is shown containing this predictor (Figure 2). 

 This has been amended for clarity: line 198-199: ‘Models were split by sex at age 14 years (male/female).’

The models were split by sex, as described in the section “Structural Equation Modelling” on line 355. Information on sex was obtained from the Recruitment Information. 

Details of the MRI protocol should be included. At the very least describing the T1 acquisition. The version of FreeSurfer is not mentioned. The supplementary tables are a very useful addition.

 The MRI scanning protocol is described elsewhere and the reference for this is provided [38].

We have included the following additional information for clarity: lines 234-243:

‘Structural MRI was performed on 3T scanners from different manufacturers [38]. A set of parameters was held constant across sites to address variations in image-acquisition techniques between scanners [38]. T1-weighted MR images were acquired using the magnetization prepared gradient echo sequence (MPRAGE) based on the Alzheimer’s Disease Neuroimaging Initiative (ADNI) protocol [38,45]. More details of the MR scanning protocol is described in depth elsewhere [38]. T1-weighted images were processed using FreeSurfer 5.3.0 to automatically parcellate the brain, including regional GMV. Amygdala GMV comprised left and right amygdala GMV, and was extracted using the Aseg Atlas [46]. The vmPFC was defined as the combination of left and right medial orbitofrontal cortex GMV, in line with previous studies (e.g. [47]), and extracted using the Desikan-Killiany Atlas [48].’

For the results, Table S1 shows all the configural family support models having large p-values, but the fit is perfect. Can the authors explain that?

 Those p-values refer to the chi-square test statistic, which tests the null hypothesis that the predicted model and observed data are equal. A large p-value (not statistically significant at the 0.05 level) for the chi-square statistic indicates that the predicted model is not significantly different from the observed data. This is the case for the Family Support model; the model fits the data, and this matches the almost-perfect CFI and RMSEA values.

It is noted that the other models presented in Table S1 (Socioeconomic Stress, Peer Problems and Emotional Symptoms) have a small and statistically significant p-value. In the main manuscript in line 446, we provide a reference that states that a statistically significant chi-square value is common is models with large sample sizes [59].

To improve clarity, in the main manuscript we have included the follow on lines 310-312: ‘A statistically significant chi-square value is common in models with large sample sizes because there is strong statistical power to detect small differences [57]. Therefore, less emphasis was placed on this statistic.’

There is a further analysis (strength of peer problem-emotional symptoms relationship) mentioned in the 2nd paragraph of the discussion, but this is not shown. This could be included in the supplementary material.

 This information is present in the Results section, under ‘Structural Equation Modelling’.

We have rearranged the information and added a title sub-section to highlight this result (lines 471-475).

‘Testing sex differences

In model 2, there was no significant difference in model fit when coefficients were constrained to equality by sex for peer problems as a predictor of emotional symptoms (χ2 = 2.284, df = 1, p = 0.131) and for socioeconomic stress as a predictor of family support (χ2 = 2.675, df = 1, p = 0.102) which suggests no sex differences in the magnitude of the relationships.’ 

We also refer to this section in the Discussion:

Lines 526-527: ‘(see Results sub-section ‘Testing sex differences’)’

The discussion needs more support for the Family Stress model. Also, more recent references might demonstrate findings similar to this study and regarding SES and mental health being ‘irrespective of gender’. 

Additional references have been included in the discussion which support the Family Stress model: line 506 ‘[4,33–36]’.

We have searched the literature and we are not aware of a recent reference that demonstrates similar findings to the study. However if the reviewer could point us to the specific reference(s) they have in mind, we would be very happy to include them.

It is stated that ‘This study goes further to show that stress from socioeconomic conditions affect whole brain structure’. This may be misleading. As only the total brain volume was used (minus the ventricles, CSF and dura) and brain subdivisions were not considered individually, I suggest to rephrase this to ‘ whole brain volume’. instead of implying that the ‘whole structure is different. 

We agree with your comments. This has been changed accordingly: lines 579-580: ‘This study goes further to show that stress from socioeconomic conditions affect whole brain volume’.

Which model is implied in ‘In this model, family support did not directly influence emotional symptoms, nor did it mediate the effect of peer problems on emotional symptoms’? The WBV correction is emphasised in the next paragraph, so I think that clarifying the model is important. 

Family support did not directly influence emotional symptoms nor mediate the effect of peer problems on emotional symptoms for both model 1 (no WBV correction) and model 2 (WBV correction). 

To avoid confusion, this has been changed to: line 586-587: ‘Family support did not directly influence emotional symptoms, nor did it mediate the effect of peer problems on emotional symptoms.’

Even though the authors have included the different centres as a covariate, I think that another limitation is lack of standardisation of the data. If I am not mistaken, the scanners and coils used were different and there is no discussion on how that affects the homogeneity of the dataset, regardless of implementing the same protocol everywhere (I assume this is the case!).

 You are correct about the same ADNI protocol being used. We have included information about the T1 acquisition based on your previous comment. 

IMAGEN recognised the issue with using multiple scanners and how it could affect the homogeneity of the dataset. This issue, and how they implemented a set of procedures to minimise variation, is discussed in detail in reference 37 (Schumann et al., 2010). First, a set of parameters that was compatible with all scanners was held constant across all sites. In terms of the coils, the best manufacturer-specific coil was used at all sites with the same scanner type. Quality control procedures were also implemented at each site, which included scanning phantoms and having volunteers scanned regularly at each site to determine variability between sites. Pre-processing was done at a central site using an automated pipeline and attempted to account for inter-site variability, such as using a template that had data from all centres.

We recognise that a lack of standardisation could still be an issue, despite the protocols by IMAGEN and the inclusion of centre site in the statistical model. 

Lines 637-643: ‘IMAGEN is a multi-centre study designed to maximise sample size. Different scanners are used at different sites for the neuroimaging assessment. To minimise variability between sites, a central protocol was used between sites and quality control and pre-processing procedures were implemented, explained in depth elsewhere [38]. Recruitment centre was included as a covariate in the analysis to further account for potential homogeneity. However, it is acknowledged that variability between sites could have affected the results in the current analysis.’

Typo: page 4: ‘This is important given retrospective reports show that half of all individuals’

 This has been changed as follows: line 61 ‘This is important as retrospective reports show that half of all individuals...’

Typo: S3: In the note. 

This has been changed to: ‘Note: Statistically significant values (p < .05) are in bold for ease of reading.’

Reviewer 2

The authors tested the hypothesis that sex determines/influences the relationships between social environment, brain structure (from MR), and emotional symptoms indicative for mental health issues. They did so in a large dataset (>2000) consisting of 14-year-olds because of the particular sensitivity to the social environment, and intense brain development at that age. The study concept is interesting and the study is mostly well-executed, but I have a number of concerns:

 We are pleased that the reviewer finds our study interesting and well executed.

1. Page 7: “a systematic review found conflicting results for sex differences between socioeconomic status and mental health difficulties” � clarify sentence; it reads as though some words are missing. 

This has been changed to ‘The existence of sex differences in the relationship between SES and mental health difficulties is debated, with a systematic review finding conflicting results [31].’ (lines 122-123).

2. Page 8: Incorporate research questions into the paragraph before (formulated in normal sentence form, not ?), instead of having a separate paragraph. What were your hypotheses?

 We have incorporated the research questions into the previous paragraph:

‘We investigated the following: how social factors interact and are associated with emotional symptoms for males and females at age 14 years, whether family support buffers against any negative effect of peer problems on mental health, how regional brain structure is associated with emotional symptoms, and whether social factors affect regional brain structure to have a cascading effect on emotional symptoms.’ (lines 150-154)

We have also included hypotheses:

‘Hypotheses

1) For social factors, peer problems and socioeconomic stress will positively predict emotional symptoms for both males and females at age 14 years. The effect size will be stronger for females compared to males due to the stronger negative effect of relational victimisation and stress on emotional symptoms. Socioeconomic stress will negatively predict family support, but there is no specific hypothesis about whether family support will directly predict emotional symptoms or not. In addition, no specific direction is predicted for the association between family support and peer problems, and thus whether family support mediates the relationship between peer problems and emotional symptoms.

2) There will be a significant association between amygdala and ventromedial prefrontal cortex (vmPFC) grey matter volume (GMV) and emotional symptoms, and this will be different between sex. Due to inconsistencies in the literature, no specific direction is predicted.

3) Social factors will be associated with brain structure; there will be a significant association between socioeconomic stress and amygdala/vmPFC GMV. Amygdala/vmPFC GMV will mediate the relationship between socioeconomic stress and emotional symptoms, with sex-specific findings predicted.’ (lines 157-174)

3. Page 9, Participants: Where was ethics approval obtained?

 The information has been included as follows: 

‘Local ethics research committees approved the study at each site (London, England: Psychiatry, Nursing and Midwifery Research Ethics Subcommittee, Waterloo Campus, King’s College London; Nottingham, England: University of Nottingham Medical School Ethics Committee; Mannheim, Germany: Medizinische Fakultaet Mannheim, Ruprecht Karl Universitaet Heidelberg and Ethik-Kommission II an der Fakultaet fuer Kliniksche Medizin Mannheim; Dresden, Germany: Ethikkommission der Medizinischen Fakultaet Carl Gustav Carus, TU Dresden Medizinische Fakultaet; Hamburg, Germany: Ethics Board, Hamburg Chamber of Physicians; Paris, France: CPP IDF VII (Comité de protection des personnes Ile de France), ID RCB: 2007-A00778-45 September 24, 2007; Dublin, Ireland: TCD School of Psychology REC; and Berlin, Germany: Ethics Committee of the Faculty of Psychology).’ (lines 187-196).

4. Page 9, Measures: How did self-identified sex correspond to self-identified gender? Gender non-conformity may have a huge impact on the studied relationships with social, brain, and emotional variables. Please comment on this in the Discussion at the very least.

 We agree that gender non-conformity may have an impact on the results of the study. Unfortunately, this dataset did not allow us to investigate this; at recruitment, participants were asked to report their sex, with only ‘male’ and ‘female’ options available. 

We have added the following to the Limitations section of the Discussion:

‘Sex differences were investigated in the current study, however we were unable to investigate the role of gender non-conformity due to this information not being available. Gender non-conformity could have influenced the study findings, due to effects on depressive symptoms and bullying victimisation [73]. Future studies could look at both sex and gender differences in the role of social and neurobiological factors in emotional symptoms.’ (lines 652-656).

5. Page 9, Measures: For the questionnaires, what is the actual variable included in the SEM? Total (sum) score? Please describe.

 We have added the following information to clarify the variables used in the SEM. Lines 200-202:

‘Separate latent variables were created for socioeconomic stress, family support, peer problems and emotional symptoms using the questionnaires and items presented in Table 1.’ 

6. Page 11, Family support: The family support is asked from the parental perspective, which may not correspond well with the child’s perspective (as alluded to in the Discussion). Are there data available on how well they correspond? Please report them. 

Perceived family support from the child’s perspective may be more predictive of mental health. The Affirmation section of Family Life Questionnaire was used to measure family support, and this is parent-reported only. Other measures of parent and child reports of family support have found discrepancies. We have included the following information in the limitations section:

‘Indeed, other measures of parent and child reports of family support have found discrepancies. Correlations between parent and child reports of parent support are weak [75], with parents reporting themselves to be more supportive compared to child reports [76]. Importantly, adolescents who reported poorer parent practices compared to parents were at higher risk of internalising symptoms [76]; this discrepancy therefore reveals information about the adult-child relationship that has implications for mental health. Unfortunately, the Family Life Questionnaire in the current study is parent-reported only, and other measures of child-reported family support were not available in the IMAGEN dataset, so this could not be explored in the current study..’ (lines 627-635)

7. Page 11, Peer problems: Specify what you mean exactly rather than saying “response format is the same” (as something explained later on).

This has been changed as follows: lines 221-222: ‘Participants responded to items such as being alone, being liked by peers, and being bullied using a three-point Likert scale.’

8. Page 12, Regional grey matter volume: Which atlas was used to extract the measures? Please specify. 

We have added the following information:

‘Amygdala GMV comprised left and right amygdala GMV, and was extracted using the Aseg Atlas [46]. The vmPFC was defined as the combination of left and right medial orbitofrontal cortex GMV, in line with previous studies (e.g. [47]), and extracted using the Desikan-Killiany Atlas [48].’ (lines 240-243)

‘WBV was defined as the ‘BrainSegVolNotVent’ variable derived from FreeSurfer using the Aseg Atlas [46].’ (lines 248-249) 

9. Page 13, Outliers: How many outliers were there for each variable, and how many were multivariate outliers? Does the latter suggest potential for bias? 

Figure 1 shows that 41 neuroimaging outliers were removed. We have included additional information in the manuscript regarding how many outliers there were for each variable and how many were multivariate outliers. There were few multivariate outliers in total and multivariate outliers followed the same direction: a high outlier in one variable predicted a high outlier in the other. Therefore, this suggests minimal potential for bias. 

We have added the following to the ‘Analysis strategy’ section: lines 284-289.

‘The number of outliers for each neuroimaging variable were as follows: WBV (n = 25), amygdala (n = 19), vmPFC (n = 21). Univariate and multivariate outliers were as follows: single variable (n = 21), WBV, amygdala and vmPFC (n = 4), amygdala and WBV (n = 3), vmPFC and WBV (n = 12), amygdala and vmPFC (n = 1). Multivariate outliers followed the same direction, i.e. if one value was 3 standard deviations below the mean, the other value also followed this.’

10. Page 13, Covariates: As I understood it, individuals received a “Yes” for the variable psychiatric diagnosis if they had any (one or more) psychiatric diagnosis, without regard for the type. This seems like a very general variable; it makes more sense to me to create a variable like this per main diagnosis, given the potential differentiated effects of various diagnoses, and the sex bias of certain diagnoses.

 We agree that there may be differences depending on disorders and that there are sex biases depending on certain disorders. The DAWBA covers many different diagnoses: anxiety disorders, mood disorders, bipolar disorders, autism spectrum disorders, eating disorders. The distribution of diagnoses is found in a later comment. We did not have information on main diagnosis; therefore, investigating the effect of different diagnoses resulted in model non-convergence due to multi-collinearity of comorbid diagnoses. Models which looked at different diagnosis groupings and which excluded any psychiatric diagnosis resulted in similar findings to the main findings. 

11. Page 13, Covariates: “Mean PDS scores were derived for males and females separately.” -> If the mean is calculated across an individual’s PDS items, it is self-evident that they are derived separately for males and females as well. Furthermore, is it customary to calculate the mean PDS score over the total PDS score?

 Different items were available to males and females for the PDS items, such as facial hair for males and menarche for females, so these were specified accordingly. Only participants who answered all questions relevant to their sex had their mean score calculated.

Both the total score and mean score can be used. We followed the approach of previous papers that used the mean score.

12. Page 14, Analysis strategy: Instead of “These cases”, do you mean “96 cases”? 

Yes – this refers to the participants without complete data in all variables used in the model, which was 96 cases. We have changed this accordingly:

Line 291: ‘Ninety-six cases’

13. Page 14, Analysis strategy: “Next, measurement invariance [analysis] and …” 

This has been changed in the text:

Line 299: ‘Next, measurement invariance analysis’

14. Page 15, Measurement invariance: the last paragraph about the numbers of response categories is not fully clear to me. Please clarify. 

This has been re-phrased:

‘Equivalence of item thresholds refer to whether the boundaries between ordinal responses of an item are similar between groups. In the threshold invariance model, item thresholds are fixed to equality between groups and model fit is compared to the configural model. In order to do this, at least three degrees of freedom are required, which refers to four ordinal response categories per item [60]. This was able to be done for the family support model, however, for the socioeconomic stress, peer problems and emotional symptoms models, items only had three response categories, therefore the fit of the threshold invariance model was equivalent to the configural model due to limited degrees of freedom. For this reason, threshold invariance was assumed between sex for the socioeconomic stress, peer problems and emotional symptoms models, and this model was considered the baseline model [60].’ (lines 324-333)

15. Page 19, Descriptive statistics: Which ‘spread’ statistic are you referring to specifically?

 This is referring to the standard deviation. This has been changed in the text:

Lines 386-387: ‘Furthermore, amygdala and vmPFC GMV had a larger standard deviation in males compared to females.’

16. Page 20, Table 2: Also report t-statistics and df, not just p-vals.

 This has been updated in Table 2 (page 23).

17. Page 21: What psychiatric diagnoses were present in the cohort, how frequently, and how did their frequencies differ among the sexes? Is there confounding possible as a result?

 The table below summarises the psychiatric diagnoses that were present in the cohort. As expected, there were more males with an ADHD/Autism diagnosis than females, and more females with a Mood or Anxiety disorder compared to males.

 DSM ICD 

Diagnosis Male n Female n Total n Male n Female n Total n

ADHD/Autism 44 12 56 35 11 46

Mood Disorder 32 96 128 33 95 128

Anxiety Disorder 17 62 79 18 62 80

Conduct/Oppositional Disorder 29 33 62 31 30 61

Other Disorder 11 23 34 12 27 39

Note: ADHD/Autism: ADHD Combined, ADHD Hyperactive, ADHD Impulsive, ADHD Other, ADHD Any, PDD/Autism; Mood Disorder: Emotional disorder, Major depression, Mania/Bipolar, Other depression; Anxiety Disorder: Agoraphobia, Generalised anxiety disorder, OCD, Other anxiety disorder, Panic disorder, PTSD, Separation anxiety, Social phobia, Specific phobia; Conduct/Oppositional Disorder: Any Conduct/Oppositional Disorder, Conduct disorder, Oppositional defiant disorder, Other disruptive disorder; Other Disorder: Other disorder, Eating disorder, Tic disorder. 

As mentioned in a previous comment, we have looked at different diagnosis groupings, including participants with a mood or anxiety disorder only, and they showed similar results to the main analysis.

18. Page 21: Most parents responded “No/NA” to socioeconomic stress items. What does this mean for your further analyses? Do you have enough non-zero socioeconomic stress data points to be able to reliably test the effect?

 For each of the socioeconomic stress indicators, non-zero data points range between 5.1% (problems with neighbours/neighbourhood - females) to 33.56% (financial difficulties - females). There does seem to be a floor effect due to the nature of the items – many families may not have problems with their neighbours or neighbourhood. The latent variable of socioeconomic stress had a good fit and measurement invariance was achieved (see S1 Appendix and S1 Table). It was found that the ‘problems with neighbours/neighbourhood’ item had a low loading, which may have been due to the low non-zero options, however it was retained in the model as removal of it resulted in worse fit.

In terms of whether the model would have been affected by indicator items that were skewed towards zero values, the WLSMV estimator and robust fit statistics were used to address this.

19. Page 21: Report t-test/chi-square test statistics for each of these group comparison statements. (t/X2, df, p) 

This has been amended in the text:

‘Responses to categorical and ordinal-level items are detailed in Table 3. A higher proportion of females had a psychiatric diagnosis compared to males (χ2 = 5.945, df = 1, p = 0.015). Recruitment was fairly distributed; Dublin had a smaller proportion and Nottingham had a larger proportion of the sample, but this was the same for both sexes (χ2 = 5.528, df = 7, p = 0.596). The majority of parents positively affirmed family support items. However, for the item “Liked and respected for who s/he is”, there was a significant sex difference (χ2 = 9.018, df = 3, p = 0.029). Parents of male adolescents were more likely to respond “A medium amount” (post-hoc residual = 2.994, p = 0.022) and less likely to respond “A great deal” (post-hoc residual = -2.811, p = 0.040) compared to parents of female adolescents. There were sex differences in responses to all emotional symptoms items (all χ2 ≥ 78.436, df = 2, ps < 0.001); males were more likely to answer “Not true” and less likely to answer “Somewhat True” and “Certainly True” (all post-hoc residuals ≥ ±2.983, ps ≤ 0.017) compared to females. Peer problems responses were mostly similar across both sexes, although the item “I have one good friend or more” was different between sex (χ2 = 10.970, df = 2, p = 0.004), with males more likely to answer “Somewhat True” (post-hoc residual = 2.877, p = 0.024) and less likely to answer “Certainly True” (post-hoc residual = -3.290, p = 0.006) compared to females. Most parents responded “No/Does not apply” to socioeconomic stress items and the distribution was similar between sexes (all χ2 ≤ 4.459, df = 2, ps ≥ 0.108).’ (lines 393-410)

For consistency, the chi-square df value has also been added to line 383:

‘(χ2 = 1.387, df = 1, p = 0.239)’ 

20. Page 22-23: Report means/SD by sex (plus difference test) for questionnaire sum scores if that is what was included in the SEMs.

 The sum scores were not included in the SEMs. Individual items were used to create latent variables, as detailed in Table 1.

21. Page 24, Measurement invariance: Describe the main take home message from these analyses here, in addition to referring to the supplement.

 We have added the following information to the Measurement Invariance results section:

‘Strict measurement invariance was achieved for parent-reported socioeconomic stress and family support, as well as child-reported peer problems and emotional symptoms. This showed that the same construct was being measured between sex and it allowed comparison of latent mean values between sex. Full results for the measurement invariance analysis are presented in S1 Appendix, and S1 and S2 Tables. There was no significant difference in the latent mean values between sex for socioeconomic stress (estimate = 0.040, SE = 0.075, p = 0.595) or family support (estimate = -0.083, SE = 0.066, p = 0.205). The mean value for males was larger for peer problems (estimate = 0.136, SE = 0.065, p = 0.036) and smaller for emotional symptoms compared to females (estimate = -0.926, SE = 0.075, p < 0.001). There were some items with low standardised loadings (< 0.50) for both sexes in the measurement invariance models – ‘problems with neighbours/neighbourhood’ for socioeconomic stress and ‘I get a lot of headaches, stomach-aches or sickness’ for emotional symptoms. Fixing the loadings of these items to zero in a separate models resulted in significantly worse model fit (socioeconomic stress: Δχ2 = 28.561, Δdf = 1, p < 0.001; emotional symptoms: (Δχ2 = 216.89, Δdf = 1, p < 0.001), therefore these items were retained in the model.’

(lines 415-429)

22. Page 25: “Socioeconomic stress was again found to be a negative predictor of emotional symptoms in males only.” -> Could this unexpected direction of effect be because the stress measure is not sensitive enough? (see related comment above) 

Thank you for the question; we agree that this was worth investigating further.

To check the validity of the latent variable of socioeconomic stress, we assessed whether it was predicted by other measures of socioeconomic status. Mother’s and father’s highest education was available in the data (8-point scale, 1 = Professional qualification e.g. PhD, MD, Master’s, 8 = None) and it was answered by the majority of participants in the sample (n = 1938). We created a parental education variable comprised of both mother’s and father’s highest education. Values were reverse-scored and summed so that a higher score indicated higher combined educational achievement. 

We found that higher parental education was associated with lower socioeconomic stress, which provides evidence for the validity of socioeconomic stress. 

We also added parental education to the main model to test whether the significant associations found related to socioeconomic stress were explained by parental education. Results were largely the same, however the significant association between socioeconomic stress and emotional symptoms for males was non-significant. This finding has been detailed in an additional section:

Lines 476-513:

‘Sensitivity analysis

To check the validity of the latent variable of socioeconomic stress, we investigated whether it was predicted by a more objective marker of socioeconomic status - parental education. The addition of parental education to the model also allowed us to test whether the significant associations found related to socioeconomic stress were explained by parental education.

Parental education was added into model 2 as a predictor of: socioeconomic stress, emotional symptoms, family support, peer problems, WBV, amygdala GMV and vmPFC GMV. We hypothesised that parental education would be negatively associated with socioeconomic stress. We also predicted that the associations of interest would remain statistically significant as in model 2 with the addition of parental education.

Parental education was comprised of both mother’s and father’s highest education (8-point scale, 1 = Professional qualification e.g. PhD, MD, Master’s, 8 = None) and the data were present for the majority of participants in the sample (n = 1938). Values were reverse-scored and summed for both mother and father so that a higher score indicated higher combined educational achievement. 

The model was a good fit to the data: robust χ2 = 1019.611, p-value < 0.001, robust CFI = 0.934, robust RMSEA = 0.024 [0.021, 0.027]. Regression results are found in STable 5. 

As predicted, higher parental education was associated with lower socioeconomic stress (male/female β = -0.250/-0.241, p < 0.001), which provides evidence for the validity of socioeconomic stress. 

The other main findings are as follows:

• Peer problems positively predicted emotional symptoms for males (β = 0.623, p < 0.001) and females (β = 0.494, p < 0.001). Parental education did not predict emotional symptoms for either sex.

• Socioeconomic stress negatively predicted family support for males (β = -0.177, p = 0.001) and females (β = -0.314, p < 0.001). Parental education positively predicted family support for females only (β = 0.118, p = 0.010).

• For females, socioeconomic stress negatively predicted whole brain volume (β = -0.105, p = 0.027). Parental education positively predicted whole brain volume for both males (β = 0.148, p < 0.001) and females (β = 0.109, p = 0.002).

• For males, vmPFC GMV negatively predicted emotional symptoms (β = -0.139, p = 0.019). Parental education did not predict vmPFC GMV.

• However, for males, socioeconomic stress no longer significantly predicted emotional symptoms (β = -0.105, p = 0.071).

The findings remained largely the same, which suggests that these effects are not due to the confounding effects of parental education. The only significant difference in results is that socioeconomic stress was no longer a statistically significant negative predictor of emotional symptoms for males.’

The following sections have also been amended, as socioeconomic stress was no longer a reliable predictor of emotional symptoms for males:

Lines 38-41:

‘ventromedial prefrontal cortex grey matter volume was negatively associated with emotional symptoms for males when corrected for whole brain volume, and socioeconomic stress was negatively associated with whole brain volume for females.’

Line 519: 

‘vmPFC GMV was…’

Lines 542-563:

‘In addition, socioeconomic stress was a negative predictor of family support, even when parental education was factored into the model. This supports the Family Stress Model, which posits that socioeconomic difficulties result in decreased parental availability and support for their children [4,33–36]. Initially in the WBV-included model, socioeconomic stress was a negative predictor of emotional symptoms in males, however this finding was non-significant when parental education was added into the model as part of the sensitivity analysis. Therefore, the relationship could be partly explained by parental education, which reflects parental status or resources. Interestingly, parental education was significantly positively associated with family support for females only. This suggests that parental education may be associated with support specific to gender-differentiated parenting practices. A meta-analysis found that parents used more autonomy-supportive strategies – which includes affirmation as used in the current study – towards girls rather than boys when looking at studies from the 1990s onwards. Before the 1990s, the effect was found in boys instead, which reflects cultural changes in parenting practices, and shows how notions of support are dependent on cultural norms [64]. Socioeconomic stress and parental education were not directly associated with emotional symptoms for males and females. This was unexpected given the wealth of research linking low socioeconomic status with poor adolescent mental health for both males and females [31,32]. Because the current study uses cross-sectional data, we are unable to determine the temporality of socioeconomic factors and family support, and possible sex differences. Future longitudinal analyses will be able to untangle these relationships and whether there is an effect on adolescent emotional symptoms.’

Lines 578-580:

‘The current study also found that parental education predicted WBV in both males and females, but this study goes further to show that stress from socioeconomic conditions affect whole brain volume’

Lines 677-678:

‘Family socioeconomic stress was related to family support and female brain volume.’

Lines 967-968:

‘S5 Table. Regression statistics for sensitivity analysis with the inclusion of parental education.’

Funding Statement:

‘JS is funded by the ESRC-BBSRC Soc-B Centre for Doctoral Training (ES/P000347/1). 

TB served in an advisory or consultancy role for ADHS digital, Infectopharm, Lundbeck, Medice, Neurim Pharmaceuticals, Oberberg GmbH, Roche, and Takeda. He received conference support or speaker’s fee by Lilly, Medice, Novartis, Shire and Takeda. He has been involved in clinical trials conducted by Shire & Viforpharma. He received royalties from Hogrefe, Kohlhammer, CIP Medien, Oxford University Press. The present work is unrelated to the above grants and relationships. 

LM is partially funded by the National Institute for Health and Care Research (NIHR) Applied Research Collaboration Greater Manchester (ARC-GM; reference: NIHR200174). The views expressed in this publication are those of the author(s) and not necessarily those of the National Institute for Health and Care Research or the Department of Health and Social Care.

The funders listed above provided support in the form of salaries for authors JS, TB and LM, but did not have any additional role in the study design, data collection and analysis, decision to publish, or preparation of the manuscript. The specific roles of these authors are articulated in the ‘author contributions’ section.

The IMAGEN study was funded by the European Union-funded FP6 Integrated Project IMAGEN (Reinforcement-related behaviour in normal brain function and psychopathology) (LSHM-CT- 2007-037286), the Horizon 2020 funded ERC Advanced Grant ‘STRATIFY’ (Brain network based stratification of reinforcement-related disorders) (695313), Human Brain Project (HBP SGA 2, 785907, and HBP SGA 3, 945539), the Medical Research Council Grant 'c-VEDA’ (Consortium on Vulnerability to Externalizing Disorders and Addictions) (MR/N000390/1), the National Institute of Health (NIH) (R01DA049238, A decentralized macro and micro gene-by-environment interaction analysis of substance use behavior and its brain biomarkers), the National Institute for Health Research (NIHR) Biomedical Research Centre at South London and Maudsley NHS Foundation Trust and King’s College London, the Bundesministeriumfür Bildung und Forschung (BMBF grants 01GS08152; 01EV0711; Forschungsnetz AERIAL 01EE1406A, 01EE1406B; Forschungsnetz IMAC-Mind 01GL1745B), the Deutsche Forschungsgemeinschaft (DFG grants SM 80/7-2, SFB 940, TRR 265, NE 1383/14-1), the Medical Research Foundation and Medical Research Council (grants MR/R00465X/1 and MR/S020306/1), the National Institutes of Health (NIH) funded ENIGMA (grants 5U54EB020403-05 and 1R56AG058854-01). Further support for the IMAGEN study was provided by grants from: - the ANR (ANR-12-SAMA-0004, AAPG2019 - GeBra), the Eranet Neuron (AF12-NEUR0008-01 - WM2NA; and ANR-18-NEUR00002-01 - ADORe), the Fondation de France (00081242), the Fondation pour la Recherche Médicale (DPA20140629802), the Mission Interministérielle de Lutte-contre-les-Drogues-et-les-Conduites-Addictives (MILDECA), the Assistance-Publique-Hôpitaux-de-Paris and INSERM (interface grant), Paris Sud University IDEX 2012, the Fondation de l’Avenir (grant AP-RM-17-013 ), the Fédération pour la Recherche sur le Cerveau; the National Institutes of Health, Science Foundation Ireland (16/ERCD/3797), U.S.A. (Axon, Testosterone and Mental Health during Adolescence; RO1 MH085772-01A1), and by NIH Consortium grant U54 EB020403, supported by a cross-NIH alliance that funds Big Data to Knowledge Centres of Excellence, the National Institute for Health Research (NIHR) Biomedical Research Centre (BRC) at South London and Maudsley NHS Foundation Trust (SLaM) and King’s College London (KCL). 

The funders had no role in study design, data collection and analysis, decision to publish, or preparation of the manuscript.’

Acknowledgements Section:

‘IMAGEN Consortium author list:

Arun L.W. Bokde7, Sylvane Desrivières8, Herta Flor5,9, Antoine Grigis10, Hugh Garavan11, Penny Gowland12, Andreas Heinz13, Rüdiger Brühl14, Jean-Luc Martinot15, Marie-Laure Paillère Martinot16, Eric Artiges17, Dimitri Papadopoulos Orfanos10, Tomáš Paus 18, 19, Luise Poustka20, Sarah Hohmann4, Sabina Millenet4, Juliane H. Fröhner21, Lauren Robinson22, Michael N. Smolka21, Henrik Walter13, Jeanne Winterer13, 23, Robert Whelan24, Gunter Schumann4, 25

7 Discipline of Psychiatry, School of Medicine and Trinity College Institute of Neuroscience, Trinity College Dublin, Dublin, Ireland; 8 Centre for Population Neuroscience and Precision Medicine (PONS), Institute of Psychiatry, Psychology & Neuroscience, SGDP Centre, King’s College London, United Kingdom; 9 Department of Psychology, School of Social Sciences, University of Mannheim, 68131 Mannheim, Germany; 10 NeuroSpin, CEA, Université Paris-Saclay, F-91191 Gif-sur-Yvette, France; 11 Departments of Psychiatry and Psychology, University of Vermont, 05405 Burlington, Vermont, USA; 12 Sir Peter Mansfield Imaging Centre School of Physics and Astronomy, University of Nottingham, University Park, Nottingham, United Kingdom; 13 Department of Psychiatry and Psychotherapy CCM, Charité – Universitätsmedizin Berlin, corporate member of Freie Universität Berlin, Humboldt-Universität zu Berlin, and Berlin Institute of Health, Berlin, Germany; 14 Physikalisch-Technische Bundesanstalt (PTB), Braunschweig and Berlin, Germany; 15 Institut National de la Santé et de la Recherche Médicale, INSERM U1299 “Developmental trajectories & psychiatry””; Université Paris-Saclay, Ecole Normale supérieure Paris-Saclay, CNRS, Centre Borelli; Gif-sur-Yvette, France; 16 Institut National de la Santé et de la Recherche Médicale, INSERM U1299 “Developmental trajectories & psychiatry””; Université Paris-Saclay, Ecole Normale supérieure Paris-Saclay, CNRS, Centre Borelli; Gif-sur-Yvette; and AP-HP.Sorbonne Université, Department of Child and Adolescent Psychiatry, Pitié-Salpêtrière Hospital, Paris, France; 17 Institut National de la Santé et de la Recherche Médicale, INSERM U1299 “Developmental trajectories & psychiatry””; Université Paris-Saclay, Ecole Normale supérieure Paris-Saclay, CNRS, Centre Borelli; Gif-sur-Yvette; and Etablissement Public de Santé (EPS) Barthélemy Durand, 91700 Sainte-Geneviève-des-Bois, France; 18 Departments of Psychiatry and Neuroscience and Centre Hospitalier Universitaire Sainte-Justine, University of Montreal, Montreal, Quebec, Canada; 19 Departments of Psychiatry and Psychology, University of Toronto, Toronto, Ontario, Canada; 20 Department of Child and Adolescent Psychiatry and Psychotherapy, University Medical Centre Göttingen, von-Siebold-Str. 5, 37075, Göttingen, Germany; 21 Department of Psychiatry and Neuroimaging Center, Technische Universität Dresden, Dresden, Germany; 22 Department of Psychological Medicine, Section for Eating Disorders, Institute of Psychiatry, Psychology and Neuroscience, King’s College London, London, SE5 8AF, UK; 23 Department of Education and Psychology, Freie Universität Berlin, Berlin, Germany; 24 School of Psychology and Global Brain Health Institute, Trinity College Dublin, Ireland; 25 PONS Research Group, Dept of Psychiatry and Psychotherapy, Campus Charite Mitte, Humboldt University, Berlin and Leibniz Institute for Neurobiology, Magdeburg, Germany, and Institute for Science and Technology of Brain-inspired Intelligence (ISTBI), Fudan University, Shanghai, P.R. China.’

Competing Interests Statement:

‘TB’s commercial affiliation does not alter our adherence to PLOS ONE policies on sharing data and materials. The remaining authors have declared that no competing interests exist.’ 

Sincerely,

Jessica Stepanous

PhD Student, Division of Neuroscience and Experimental Psychology

Zochonis Building, University of Manchester, Manchester, UK

---

## [Decision Letter · Decision Letter 1]

28 Aug 2022

PONE-D-21-30637R1Social environment and brain structure in adolescent mental health: A cross-sectional structural equation modelling study using IMAGEN dataPLOS ONE

Dear Dr. Stepanous,

Thank you for submitting your manuscript to PLOS ONE. After careful consideration, we feel that it needs still some minor changes before it is acceptable for PlosOne. Therefore, we invite you to submit a revised version of the manuscript that addresses the points raised during the review process.

We look forward to receiving your revised manuscript.

Kind regards,

Therese van Amelsvoort

Academic Editor

PLOS ONE

Journal Requirements:

Reviewers' comments:

Reviewer's Responses to Questions

**Comments to the Author**

1. If the authors have adequately addressed your comments raised in a previous round of review and you feel that this manuscript is now acceptable for publication, you may indicate that here to bypass the “Comments to the Author” section, enter your conflict of interest statement in the “Confidential to Editor” section, and submit your "Accept" recommendation.

Reviewer #1: All comments have been addressed

Reviewer #2: (No Response)

2. Is the manuscript technically sound, and do the data support the conclusions?

Reviewer #1: Yes

Reviewer #2: Yes

3. Has the statistical analysis been performed appropriately and rigorously? 

Reviewer #1: Yes

Reviewer #2: Yes

4. Have the authors made all data underlying the findings in their manuscript fully available?

Reviewer #1: Yes

Reviewer #2: No

5. Is the manuscript presented in an intelligible fashion and written in standard English?

Reviewer #1: Yes

Reviewer #2: Yes

6. Review Comments to the Author

Reviewer #1: the authors convincingly addressed my comments / the authors convincingly addressed my comments / the authors convincingly addressed my comments /

Reviewer #2: All comments have been addressed, but the response has not been directly incorporated in the manuscript itself for the following comments:

10. Page 13, Covariates: As I understood it, individuals received a “Yes” for the

variable psychiatric diagnosis if they had any (one or more) psychiatric diagnosis,

without regard for the type. This seems like a very general variable; it makes more

sense to me to create a variable like this per main diagnosis, given the potential

differentiated effects of various diagnoses, and the sex bias of certain diagnoses.

Response: We agree that there may be differences depending on disorders and that there are sex

biases depending on certain disorders. The DAWBA covers many different diagnoses:

anxiety disorders, mood disorders, bipolar disorders, autism spectrum disorders, eating

disorders. The distribution of diagnoses is found in a later comment. We did not have

information on main diagnosis; therefore, investigating the effect of different diagnoses

resulted in model non-convergence due to multi-collinearity of comorbid diagnoses.

Models which looked at different diagnosis groupings and which excluded any

psychiatric diagnosis resulted in similar findings to the main findings.

Reply: Please include this information in the Supplementary Materials (with a reference in the main text).

11. Page 13, Covariates: “Mean PDS scores were derived for males and females

separately.” -> If the mean is calculated across an individual’s PDS items, it is selfevident

that they are derived separately for males and females as well. Furthermore, is

it customary to calculate the mean PDS score over the total PDS score?

Response: Different items were available to males and females for the PDS items, such as facial

hair for males and menarche for females, so these were specified accordingly. Only

participants who answered all questions relevant to their sex had their mean score

calculated.

Reply: Please include this information in the Methods.

17. Page 21: What psychiatric diagnoses were present in the cohort, how frequently,

and how did their frequencies differ among the sexes? Is there confounding possible

as a result?

Response: The table below summarises the psychiatric diagnoses that were present in the cohort.

As expected, there were more males with an ADHD/Autism diagnosis than females,

and more females with a Mood or Anxiety disorder compared to males.

DSMICD

DiagnosisMale nFemale nTotal nMale nFemale nTotal n

ADHD/Autism441256351146

Mood Disorder32961283395128

Anxiety Disorder176279186280

Conduct/Oppositional Disorder293362313061

Other Disorder112334122739

Note: ADHD/Autism: ADHD Combined, ADHD Hyperactive, ADHD Impulsive, ADHD

Other, ADHD Any, PDD/Autism; Mood Disorder: Emotional disorder, Major depression,

Mania/Bipolar, Other depression; Anxiety Disorder: Agoraphobia, Generalised anxiety

disorder, OCD, Other anxiety disorder, Panic disorder, PTSD, Separation anxiety,

Social phobia, Specific phobia; Conduct/Oppositional Disorder: Any

Conduct/Oppositional Disorder, Conduct disorder, Oppositional defiant disorder, Other

disruptive disorder; Other Disorder: Other disorder, Eating disorder, Tic disorder.

As mentioned in a previous comment, we have looked at different diagnosis groupings,

including participants with a mood or anxiety disorder only, and they showed similar

results to the main analysis.

Reply: Please include this table and other information in the Supplementary Materials (with a reference in the main text).

18. Page 21: Most parents responded “No/NA” to socioeconomic stress items. What

does this mean for your further analyses? Do you have enough non-zero

socioeconomic stress data points to be able to reliably test the effect?

Response: For each of the socioeconomic stress indicators, non-zero data points range between

5.1% (problems with neighbours/neighbourhood - females) to 33.56% (financial

difficulties - females). There does seem to be a floor effect due to the nature of the

items – many families may not have problems with their neighbours or neighbourhood.

The latent variable of socioeconomic stress had a good fit and measurement

invariance was achieved (see S1 Appendix and S1 Table). It was found that the

‘problems with neighbours/neighbourhood’ item had a low loading, which may have

been due to the low non-zero options, however it was retained in the model as removal

of it resulted in worse fit.

In terms of whether the model would have been affected by indicator items that were

skewed towards zero values, the WLSMV estimator and robust fit statistics were used

to address this.

Reply: Please include this information in the Supplementary Materials (with a reference in the main text).

7. PLOS authors have the option to publish the peer review history of their article (what does this mean?). If published, this will include your full peer review and any attached files.

Reviewer #1: No

Reviewer #2: No

---

## [Author Response · Author response to Decision Letter 1]

12 Sep 2022

Reviewer 2 original comments, responses, and new comments:

10. Page 13, Covariates: As I understood it, individuals received a “Yes” for the variable psychiatric diagnosis if they had any (one or more) psychiatric diagnosis, without regard for the type. This seems like a very general variable; it makes more sense to me to create a variable like this per main diagnosis, given the potential differentiated effects of various diagnoses, and the sex bias of certain diagnoses.

Response: We agree that there may be differences depending on disorders and that there are sex biases depending on certain disorders. The DAWBA covers many different diagnoses: anxiety disorders, mood disorders, bipolar disorders, autism spectrum disorders, eating disorders. The distribution of diagnoses is found in a later comment. We did not have information on main diagnosis; therefore, investigating the effect of different diagnoses resulted in model non-convergence due to multi-collinearity of comorbid diagnoses.

Models which looked at different diagnosis groupings and which excluded any psychiatric diagnosis resulted in similar findings to the main findings.

Reply: Please include this information in the Supplementary Materials (with a reference in the main text).

New response:

We have included the following supplementary tables with this information: S6 Table. Distribution of psychiatric diagnoses by sex, separately for DSM-IV and ICD-10, S7 Table. Regression statistics for the sensitivity analysis of the exclusion of participants with a psychiatric diagnosis, and S8 Table. Regression statistics for sensitivity analysis of the inclusion of mood or anxiety disorder instead of any psychiatric disorder.

We have included the following information in the main text:

Lines 520-537:

Psychiatric diagnosis

Psychiatric diagnosis was included as a covariate in the study, but sex biases in the frequencies of psychiatric disorders may have influenced the findings. The distribution of psychiatric diagnoses by sex are presented in S6 Table. There were more males with an ADHD/Autism diagnosis than females, and more females with a mood or anxiety disorder compared to males. Information on main diagnosis was not available, so investigating the effect of dummy-coded diagnoses in the same model resulted in model non-convergence due to multi-collinearity of comorbid diagnoses. Instead, we ran two additional models: one that excluded participants with any psychiatric diagnosis (see S7 Table for regression output) and one that only investigated mood or anxiety disorder diagnosis instead of any psychiatric diagnosis (see S8 Table), due to their high likelihood of comorbidity and given the focus on emotional symptoms in the current study. 

Both models showed good fit to the data. For the psychiatric diagnosis excluded model in S7 Table, there were zero responses for males for the “Not True” option for the “Gets love and affection” item in the Family Life Questionnaire, therefore the responses to “Not True” and “Somewhat True” were merged in this model. In both models, main associations of interest found in previous models remained statistically significant. Additionally, family support was negatively associated with emotional symptoms in females only in both models.

We have also included more information in the Discussion about the finding that family support was negatively associated with emotional symptoms for females in the no psychiatric diagnosis and mood or anxiety disorder only models.

Lines 546-549:

Family support was negatively associated with emotional symptoms in females only in the sensitivity analysis, where models either did not include participants with a psychiatric diagnosis or only included participants with mood or anxiety disorders.

Lines 612-626:

Family support did not directly influence emotional symptoms in models that controlled for any psychiatric diagnosis, nor did it mediate the effect of peer problems on emotional symptoms in any model. In the sensitivity analysis, models that either did not include participants with a psychiatric diagnosis or only included participants with mood or anxiety disorders found that family support was negatively associated with emotional symptoms in females only. This suggests that the link between family support and emotional symptoms in females was previously obscured by the inclusion of participants who had psychiatric diagnoses other than mood or anxiety disorders. These findings contradict previous research that found that, similarly for both sexes, family support independently predicts mental health outcomes [23] and buffers against the effect of peer problems on mental health [28,29]. Females may be more sensitive to general family support, or it may be that the type of support needs to be targeted to the problem, in order for it to have an effect. Successful social support has been found to depend on the source, type, and timing of the support [71], suggesting that general measures of family support may not be sensitive to determine a buffering effect for both sexes.

Reviewer 2 original comments, responses, and new comments:

11. Page 13, Covariates: “Mean PDS scores were derived for males and females separately.” -> If the mean is calculated across an individual’s PDS items, it is selfevident that they are derived separately for males and females as well. Furthermore, is it customary to calculate the mean PDS score over the total PDS score?

Response: Different items were available to males and females for the PDS items, such as facial hair for males and menarche for females, so these were specified accordingly. Only participants who answered all questions relevant to their sex had their mean score calculated.

Reply: Please include this information in the Methods.

New response:

This has been included in the Methods, under the Covariates section:

Lines 274-276: Different items were available to males and females for the PDS items, such as facial hair for males and menarche for females, so these were specified accordingly. Only participants who answered all questions relevant to their sex had their mean score calculated.

Reviewer 2 original comments, responses, and new comments:

17. Page 21: What psychiatric diagnoses were present in the cohort, how frequently, and how did their frequencies differ among the sexes? Is there confounding possible as a result?

Response: The table below summarises the psychiatric diagnoses that were present in the cohort. 

[S6 Table here]

As expected, there were more males with an ADHD/Autism diagnosis than females, and more females with a Mood or Anxiety disorder compared to males.

As mentioned in a previous comment, we have looked at different diagnosis groupings, including participants with a mood or anxiety disorder only, and they showed similar results to the main analysis.

Reply: Please include this table and other information in the Supplementary Materials (with a reference in the main text).

New response:

We have included S6 Table in the Supporting Information and referred to it in the main text.

Reviewer 2 original comments, responses, and new comments:

18. Page 21: Most parents responded “No/NA” to socioeconomic stress items. What does this mean for your further analyses? Do you have enough non-zero socioeconomic stress data points to be able to reliably test the effect?

Response: For each of the socioeconomic stress indicators, non-zero data points range between 5.1% (problems with neighbours/neighbourhood - females) to 33.56% (financial difficulties - females). There does seem to be a floor effect due to the nature of the items – many families may not have problems with their neighbours or neighbourhood. The latent variable of socioeconomic stress had a good fit and measurement invariance was achieved (see S1 Appendix and S1 Table). It was found that the ‘problems with neighbours/neighbourhood’ item had a low loading, which may have been due to the low non-zero options, however it was retained in the model as removal of it resulted in worse fit. In terms of whether the model would have been affected by indicator items that were skewed towards zero values, the WLSMV estimator and robust fit statistics were used to address this.

Reply: Please include this information in the Supplementary Materials (with a reference in the main text).

New response:

The following information has been added to the S1 Appendix:

For each of the socioeconomic stress indicators, non-zero data points ranged between 5.1% (problems with neighbours/neighbourhood - females) to 33.56% (financial difficulties - females). There appeared to be a floor effect due to the nature of the items – for example, many families may not have had problems with their neighbours or neighbourhood. Nonetheless, the latent variable of socioeconomic stress had a good fit and measurement invariance was achieved (see S1 Table). It was found that the ‘problems with neighbours/neighbourhood’ item had a low loading, which may have been due to the low non-zero options. However, it was retained in the model as removal of it resulted in worse fit. In terms of whether the model was affected by indicator items that were skewed towards zero values, the WLSMV estimator and robust fit statistics were used to address this.

The following has also been added to the main text:

Lines 432-434:

Additional information on the potential impact of the number of non-zero data points for the socioeconomic stress latent variable is described in S1 Appendix.

---

## [Decision Letter · Decision Letter 2]

2 Oct 2022

PONE-D-21-30637R2Social environment and brain structure in adolescent mental health: A cross-sectional structural equation modelling study using IMAGEN dataPLOS ONE

Dear Dr. Stepanous, Thank you for submitting your manuscript to PLOS ONE. After careful consideration, we feel that it has merit but does not fully meet PLOS ONE’s publication criteria as it currently stands. Therefore, we invite you to submit a revised version of the manuscript that addresses the points raised during the review process.

Please check my comments below.

We look forward to receiving your revised manuscript.

Kind regards,

Thiago Fernandes, MS, EbS, Sp. Neuro, PhD

Academic Editor

PLOS ONE

Journal Requirements:

Additional Editor Comments:

Thank you for your submission. You’ll notice that the concerns have been addressed.

Based on my own reading, I just suggest the authors to:

1. Double check grammar again

2. Make sure that all necessary files are on OSF

3. Avoid the excessive lengthy paragraphs

4. Report all stats parameters (assumptions check, effect sizes and CIs)

5. Provide a significance statement at the very end of the discussion. Don’t worry, it can be simple: limitations WITH further recommendations for researchers or next studies & the strengths of your findings. A smoother conclusion is also interesting

Please do apologize delay in return - we have some issues securing reviewers.

Reviewers' comments:

Reviewer's Responses to Questions

**Comments to the Author**

1. If the authors have adequately addressed your comments raised in a previous round of review and you feel that this manuscript is now acceptable for publication, you may indicate that here to bypass the “Comments to the Author” section, enter your conflict of interest statement in the “Confidential to Editor” section, and submit your "Accept" recommendation.

Reviewer #1: All comments have been addressed

2. Is the manuscript technically sound, and do the data support the conclusions?

Reviewer #1: (No Response)

3. Has the statistical analysis been performed appropriately and rigorously? 

Reviewer #1: (No Response)

4. Have the authors made all data underlying the findings in their manuscript fully available?

Reviewer #1: (No Response)

5. Is the manuscript presented in an intelligible fashion and written in standard English?

Reviewer #1: (No Response)

6. Review Comments to the Author

Reviewer #1: (No Response)

7. PLOS authors have the option to publish the peer review history of their article (what does this mean?). If published, this will include your full peer review and any attached files.

Reviewer #1: **Yes: **Neeltje van Haren

---

## [Author Response · Author response to Decision Letter 2]

12 Oct 2022

The reviewers comments have been previously addressed. The current revisions are in response to the Academic Editor.

---

## [Editor Report · Decision Letter 3]

20 Dec 2022

Social environment and brain structure in adolescent mental health: A cross-sectional structural equation modelling study using IMAGEN data

PONE-D-21-30637R3

Dear Dr. Stepanous,

Thank you.

After re-reading, the concerns were properly addressed.

Wishing you success with this study.

We’re pleased to inform you that your manuscript has been judged scientifically suitable for publication and will be formally accepted for publication once it meets all outstanding technical requirements.

Kind regards,

Thiago Fernandes, MD, EbS, Sp. Neuro, PhD

Academic Editor

PLOS ONE
---

## [Editor Report · Acceptance letter]

26 Dec 2022

PONE-D-21-30637R3 

Social environment and brain structure in adolescent mental health: A cross-sectional structural equation modelling study using IMAGEN data 

Dear Dr. Stepanous:

I'm pleased to inform you that your manuscript has been deemed suitable for publication in PLOS ONE. Congratulations! Your manuscript is now with our production department. 

Kind regards, 

on behalf of

Dr. Thiago P. Fernandes 

Academic Editor

PLOS ONE